# Ultrafast adaptive optics for imaging the living human eye

Yan Liu [1] ✉, James A. Crowell[1], Kazuhiro Kurokawa [1,2,6], Marcel T. Bernucci[1,6], Qiuzhi Ji[1,6], Ayoub Lassoued[1,3], Hae Won Jung [1,4], Matthew J. Keller[1], Mary E. Marte[1,5] & Donald T. Miller [1] ✉

Adaptive optics (AO) is a powerful method for correcting dynamic aberrations in numerous applications. When applied to the eye, it enables cellular-resolution retinal imaging and enhanced visual performance and stimulation. Most ophthalmic AO systems correct dynamic aberrations up to 1–2 Hz, the commonly-known cutoff frequency for correcting ocular aberrations. However, this frequency may be grossly underestimated for more clinically relevant scenarios where the medical impact of AO will be greatest. Unfortunately, little is known about the aberration dynamics in these scenarios. A major bottleneck has been the lack of sufficiently fast AO systems to measure and correct them. We develop an ultrafast ophthalmic AO system that increases AO bandwidth by ~30× and improves aberration power rejection magnitude by 500×. We demonstrate that this much faster ophthalmic AO is possible without sacrificing other system performances. We find that the discontinuous-exposure AO-control scheme runs 32% slower yet achieves 53% larger AO bandwidth than the commonly used continuous-exposure scheme. Using the ultrafast system, we characterize ocular aberration dynamics in six clinically-relevant scenarios and find their power spectra to be 10–100× larger than normal. We show that ultrafast AO substantially improves aberration correction and retinal imaging performance in these scenarios compared with conventional AO.

Adaptive optics (AO) is an electro-optical approach for measuring and correcting dynamic wavefront aberrations that are found in numerous applications[1]. In its most common form, AO consists of a wavefront sensor for measuring aberrations, a wavefront corrector for correcting them, and a control system that adjusts the wavefront corrector based on the measurements. Applied to the eye for the first time in 1997[2,3], AO systems are now increasingly used in ophthalmology and vision science to correct the unique optical defects (wave aberrations) in the cornea and crystalline lens of each individual eye. This application has

proven highly successful. When used in ophthalmoscopes, it enables the acquisition of the sharpest retinal images in vivo, allowing for single-cell imaging[1–19]. In vision correction devices, AO enables the sharpest images to form on the retina, leading to improved visual performance and stimulation[2,20–25].

It is well known that the aberrations of the eye vary over time, due to dynamics in the optics of the eye, tear film instabilities, and eye movement[26–31]. This variation necessitates AO with sufficient temporal bandwidth and sufficiently low latency to promptly track and correct

[1]School of Optometry, Indiana University, Bloomington, IN, USA. [2]Present address: Discoveries in Sight Research Laboratories, Devers Eye Institute, Legacy Research Institute, Legacy Health, Portland, OR, USA. [3]Present address: Centre Hospitalier National d'Ophtalmologie des Quinze-Vingts Centre d'investigation clinique, Paris, Île-de-France, France; Institut de la vision, Paris, Île-de-France, Paris, France. [4]Present address: University of Houston, Houston, TX, USA. [5]Present address: Richard L. Roudebush VAMC, Indianapolis, IN, USA. [6]These authors contributed equally: Kazuhiro Kurokawa, Marcel T. Bernucci, Qiuzhi Ji. ✉e-mail: yl144@iu.edu; dtmiller@iu.edu

these ocular aberration changes. It has been established more than two decades ago that AO systems with bandwidths of 1–2 Hz are sufficient to correct ocular aberrations[26], and this criterion has guided the design of ophthalmic AO systems since. Most systems use wavefront sensor integration (or exposure) times of 10–60 ms with AO loop rates (frequency at which an AO system updates its correction) typically not exceeding 30 Hz, resulting in AO bandwidths (range of temporal frequencies over which an AO system effectively corrects aberrations) of ~1.4 Hz[32–43]. However, the established 1–2 Hz bandwidth criterion is based on measurements acquired from healthy subjects under largely ideal conditions, which may not capture the full range of real-life scenarios encountered in clinical settings, where AO is used to image a more diverse range of eyes under more challenging conditions and more stringent time constraints. Many of these scenarios could induce additional high- and low- temporal-frequency aberrations (as described below), yet almost nothing is known about the temporal properties of ocular aberrations in these situations—a major gap given that at least half of current AO publications involve clinical research[3].

There are no established criteria for clinical AO operation[4]. Nevertheless, there is a general understanding in the field that the AO system needs to be fast enough to permit clinicians and technicians to acquire high-resolution images easily, efficiently, and robustly under all kinds of eye conditions, similar requirements as those for operation of non-AO commercial ophthalmoscopes[44]. Specifically, a clinical AO system should: (1) quickly align to the patient; (2) allow rapid focus through and about the retina (traversing distances larger than the ocular isoplanatic patch and system's depth of focus); (3) stabilize focus at a retinal depth of interest; (4) be robust against tear film disruptions, eye blinks, and eye and ophthalmic appliance (e.g., contact lens) motion; and (5) correct accommodation in non-cyclopleged eyes or in cyclopleged eyes but before or after peak cycloplegic effectiveness. Ensuring these five capabilities is more difficult in the presence of disease and aging, which could exacerbate conditions necessitating higher AO speeds. Disease and aging increase the incidence of dry eye[45] and nystagmus[46], reduce fixation stability[47,48], prohibit cycloplegia in some patients (e.g., those with narrow-angle glaucoma)[49], and can cause abnormal optics requiring correction by contact lens (e.g., keratoconus[50] and high myopia), which can move about on the eye especially after a blink. Older eyes also exhibit increased aberrations[51], compounding the temporal effect of these conditions. These conditions likely necessitate higher AO speeds, but to what extent is unknown.

The need for faster AO, even in healthy subjects, is supported by results from a recent wavefront-aberration study (with no wavefront corrector) of 50 non-cyclopleged eyes with 5-mm pupils in 23–38-year-old subjects; the study predicts that the AO loop rate needs to be ~70 Hz (when the loop gain is 0.5) in order to achieve diffraction-limited performance in 80% of the population[27]. Several high-speed ophthalmic AO systems with loop rates at or exceeding 30 Hz have been developed and have reported improved correction performance or retinal image quality[36,52–57]. However, the largest AO bandwidth reported to date of an AO system with diffraction-limited capability over a large eye pupil (≥5 mm diameter) is 4 Hz[55]. Given this moderate improvement (from 1–2 Hz to 4 Hz), it would not be surprising if these higher-speed systems lack the capability to properly measure and correct the range of dynamic aberrations that may be present in the aforementioned clinical conditions. One could imagine leveraging astronomical AO systems as they operate at much higher loop rates (kHz)[1]. Unfortunately, their wavefront sensors have insufficient pixel counts per sub-aperture (lenslet) to support the large dynamic range required for ophthalmology and vision science[1], making these systems unsuitable for use in the eye.

Here, we conduct a series of fundamental studies on the development and characterization of an ultrafast ophthalmic AO system (38.0 Hz bandwidth), its comparison to conventional ophthalmic AO, and its application in measuring and correcting ocular aberrations. This includes theoretical and experimental assessments of two fundamental AO control schemes and the impact of deformable mirror (DM) actuation, as well as predictions and characterizations of temporal and noise performances of the AO system. Because the temporal performance of our system vastly surpasses that of any ophthalmic system that we know of, it allows us to better characterize and correct the temporal spectrum of ocular aberrations in a wider range of real-life scenarios that are encountered in clinical settings. We substantiate these findings by acquiring AO retinal images in several targeted scenarios that we refer to as "clinically relevant" because they are all applicable to clinic use, either because they include a disease or eye condition or because the imaging protocol improves ease of use, efficiency of imaging, or robustness of operation. We find that ultrafast AO significantly improves image quality (sharpness of retinal cells) and reduces wavefront error compared to conventional AO. The theoretical and experimental methods we present have considerable generality for design optimization of future ultrafast AO systems.

Taken together, ultrafast ophthalmic AO offers a unique capability for visualizing cells in eyes under conditions that are more challenging to image and more likely to be encountered in the clinic. It also offers considerable potential to generate exquisitely sharp images at the retina to enhance the study of fundamental properties of visual performance under various pathological and physiologic conditions.

## Results

In the results presented in the following 10 sub-sections, we:

1. Developed an ultrafast ophthalmic AO system whose key features include a large-pixel-count, low-latency wavefront sensor and a discontinuous-exposure AO control scheme that maximizes AO bandwidth;
2. Conducted an experimental and theoretical comparison of the temporal performances of two fundamental AO control schemes, revealing that the discontinuous-exposure scheme significantly outperforms the continuous-exposure scheme for ophthalmic use;
3. Identified and corrected a performance-degrading effect of DM actuation;
4-5. Evaluated the temporal and noise performances of our ultrafast AO system using control theory and laboratory measurements, revealing that the ultrafast AO bandwidth is ~30× larger than that of conventional ophthalmic AO systems and that RMS wavefront error due to wavefront sensing noise is better than the diffraction limit;
6. Employed the wavefront sensor of our ultrafast AO system to characterize the temporal content of ocular aberrations at 342 Hz in 24 different subjects (35 subject measurements) exemplifying six clinically relevant scenarios and one control scenario. We used this information to predict the AO system speed required to achieve diffraction-limited performance;
7-10. Utilized our ultrafast AO to correct ocular aberrations in the same subjects and scenarios as in 6, demonstrating a significant improvement in aberration correction and retinal image quality over conventional AO.

### Our ultrafast ophthalmic AO system

We developed an ultrafast ophthalmic AO system based on the following key components: (1) a high-speed, low-latency Shack-Hartmann wavefront sensor (SHWS) with high spatial sampling and dynamic range; (2) highly-efficient software that minimizes the sensor data processing time; (3) a discontinuous-exposure AO operational scheme that achieves a larger AO bandwidth than the continuous-exposure scheme used in astronomy and by many ophthalmic AO groups (including us before this work); and (4) a method of minimizing the deformable mirror (DM) actuation effect while maximizing AO

**Table 1 | Summary of the key speed-related performance parameters of ultrafast ophthalmic AO and comparison with reported ophthalmic AO systems**

| Speed related parameters | Ultrafast AO | Conventional AO |
|---|---|---|
| SHWS integration time | 0.126 ms | 10–60 ms[32–34,36,37,55] |
| Data processing time | 0.5 ms | ≥6.5 ms[54] |
| Latency[a] | 3.1 ms | ≥20 ms[32,33,36,53,55] |
| Time to reach diffraction limit | | |
| model eye | 4.3 ms | 100 ms[54] |
| living human eye | 4.3–8.6 ms | ≥200 ms[36,40,53,73,74,82] |
| AO bandwidth | 38.0 Hz | ≤4 Hz[55] |
| | | ≤1.4 Hz for most systems[32–37] |

[a]Latency is defined as the time delay between the average exposure start time across rows of the SHWS camera and completion of the DM update[83,84], indicated as $T_{latency}$ in the timing diagram in Supplementary Fig. 1.

temporal performance. This section describes the first two items, while the remaining two are discussed in the two following sections.

We maximized the SHWS speed using off-the-shelf components without sacrificing the high spatial performance of our previous AO system[58,59], which featured a 97-actuator DM (DM97-15 high speed, ALPAO). We also maintained sufficient signal-to-noise ratio (SNR) of the SHWS to ensure that photon and camera noise did not affect AO performance (discussed later). The same beam power used by our previous system illuminated the eye (~420 μW at 790 nm) and the same fraction (~10%) deflected into the SHWS. The SHWS comprised a 20 × 20 microlens array (square microlenses, pitch = 0.5 mm, $f$ = 13.9 mm, SUSS MicroOptics) that sampled a 6.7 mm eye pupil (with 300 microlenses) and a high-speed streaming camera (ORCA-Lightning, Hamamatsu). The camera used rolling-shutter mode, which allowed a higher AO loop rate and sensitivity than global shutter mode (see Supplementary Note 1 for a comparison of these two modes for AO ophthalmoscopy). To minimize latency, the wavefront sensor integration time was set to 0.126 ms, 100–500× shorter than those typically used in the ophthalmic AO field (10–60 ms)[32–34,36,37,55]. The wavefront sensor camera had a readout speed of 1.445 × 10$^9$ pixels/s, 3–100× higher than that used in the field[32–34,36,37,55]. As a result, the camera achieved a maximum frame rate of 342 Hz for our area of interest of 1920 H × 1840 V pixels, more than 30× faster than the most widely used ophthalmic AO systems in academia and industry[32,33,38]. Each sub-aperture image was sampled by 45 × 45 super pixels (after 2 × 2 pixel binning). The pixel readout process had an RMS readout noise of 2e⁻, and each pixel intensity was digitized to 12 bits. To facilitate fast image data transfer to a computer, the camera used a CoaXPress 6.25 Gbps × 4 lanes interface with four cables.

Our customized AO control software ran on an off-the-shelf workstation (Precision 5820, Dell) with a 12-core CPU (Core i9-9920X, Intel). The main program was written in Python/NumPy, and device drivers, image unpacking (from three bytes per two pixels to two bytes per pixel), and centroiding algorithm was implemented in C/C++ and parallelized with OpenMP. The software completed data processing in 0.5 ms, including unpacking the 12-bit SHWS image to 16 bits, centroiding the spots and computing their displacements, detecting and handling of eye blinks, computing Zernike coefficients and root-mean-square (RMS) wavefront error, and generating voltage commands for the DM. Our software is at least 13× faster than the fastest reported ophthalmic AO system[53]. See Methods for more details and Table 1 (column 2) for a summary of our measurements of key speed-related performance parameters of our ultrafast ophthalmic AO system.

## Discontinuous-exposure scheme outperforms continuous-exposure scheme for ophthalmic use, experimentally and theoretically

To control an AO system (Fig. 1a), one can use either of the two operational schemes shown in Fig. 1b. In the continuous-exposure scheme (with the camera in internal-trigger mode), the camera exposes frames one after another continuously independent of data processing and DM actuation. In the discontinuous-exposure scheme, the control program triggers each camera exposure—using the camera's external trigger mode—only after completion of data processing and DM actuation from the previous exposure. The latencies of these two schemes are the same (Fig. 1b); however, the AO loop rate of the continuous-exposure scheme (which is equal to the SHWS frame rate) is higher, because it does not have to wait for data processing and DM actuation to be completed as required by discontinuous exposure. For our system, the AO loop rate of the continuous-exposure scheme is 342 Hz, 47% higher than that of the discontinuous-exposure scheme (233 Hz). Continuous exposure is commonly used in astronomy[60–62] and ophthalmology/vision science (including our group before this work)[35–37,40,53–55,63,64], in large part because it captures more photons in the same time interval (less dead time). However, as we will demonstrate, the discontinuous-exposure scheme permits the use of a higher AO loop gain. Although discontinuous exposure has been used by some ophthalmic AO groups due to its more straightforward implementation, to the best of our knowledge, the use of this scheme has not been described in the AO literature and before this work it was unclear which operational scheme would yield better performance for ophthalmic AO. Hence, we studied this problem first by comparing these two schemes experimentally and theoretically in order to maximize AO performance.

We characterized the temporal performance of our AO system by measuring its power rejection curve, which shows power rejection magnitude as a function of temporal frequency[1,18,35,55,60]. Power rejection magnitude is defined as the ratio of wavefront aberration power spectral densities with and without closed-loop AO correction. Hence, the lower the power rejection magnitude, the better the AO system's ability to correct ocular aberrations. Power rejection magnitude is generally small at low frequencies and initially increases with increasing frequency. The temporal frequency at which the power rejection magnitude first reaches 1 defines the temporal bandwidth of the AO system[1,18,35,55,60], and it represents the maximum temporal frequency for which the AO system can correct aberrations. In control theory, the power rejection curve is also equal to the modulus squared of the rejection transfer function (or error transfer function)[35,55,60].

We measured the power rejection curve and bandwidth of our system under each of the two fundamental AO operational schemes, using a model eye and applying pseudo-random aberrations to the DM in the system (see Methods). For this purpose, we first needed to empirically determine the optimal loop gain: one that is high enough for fast convergence yet not too high to destabilize the AO loop. Figure 1c, d shows RMS wavefront error as a function of time before and after AO was turned on at time 0 s for both continuous and discontinuous exposure and for values of loop gain straddling optimal performance. As shown in Fig. 1c for the continuous-exposure scheme, a gain of 0.55 causes oscillations or overshoots, whereas with a gain of 0.35 the oscillation is eliminated but convergence is slower; the time to reach the diffraction limit (when RMS wavefront error ≤$\lambda$/14, where $\lambda$ = 0.79 μm is the wavelength) increases by ~34%. We find that a loop gain of 0.45 is optimal for this operational scheme, yielding stable and fast AO. Using the discontinuous-exposure scheme (Fig. 1d), on the other hand, the optimal loop gain is found to be 1.0 and RMS wavefront error reaches a diffraction limit 49% faster than under the continuous-exposure scheme (4.3 ms vs. 8.4 ms), even though the loop rate of discontinuous exposure is 32% slower (233 Hz vs. 342 Hz).

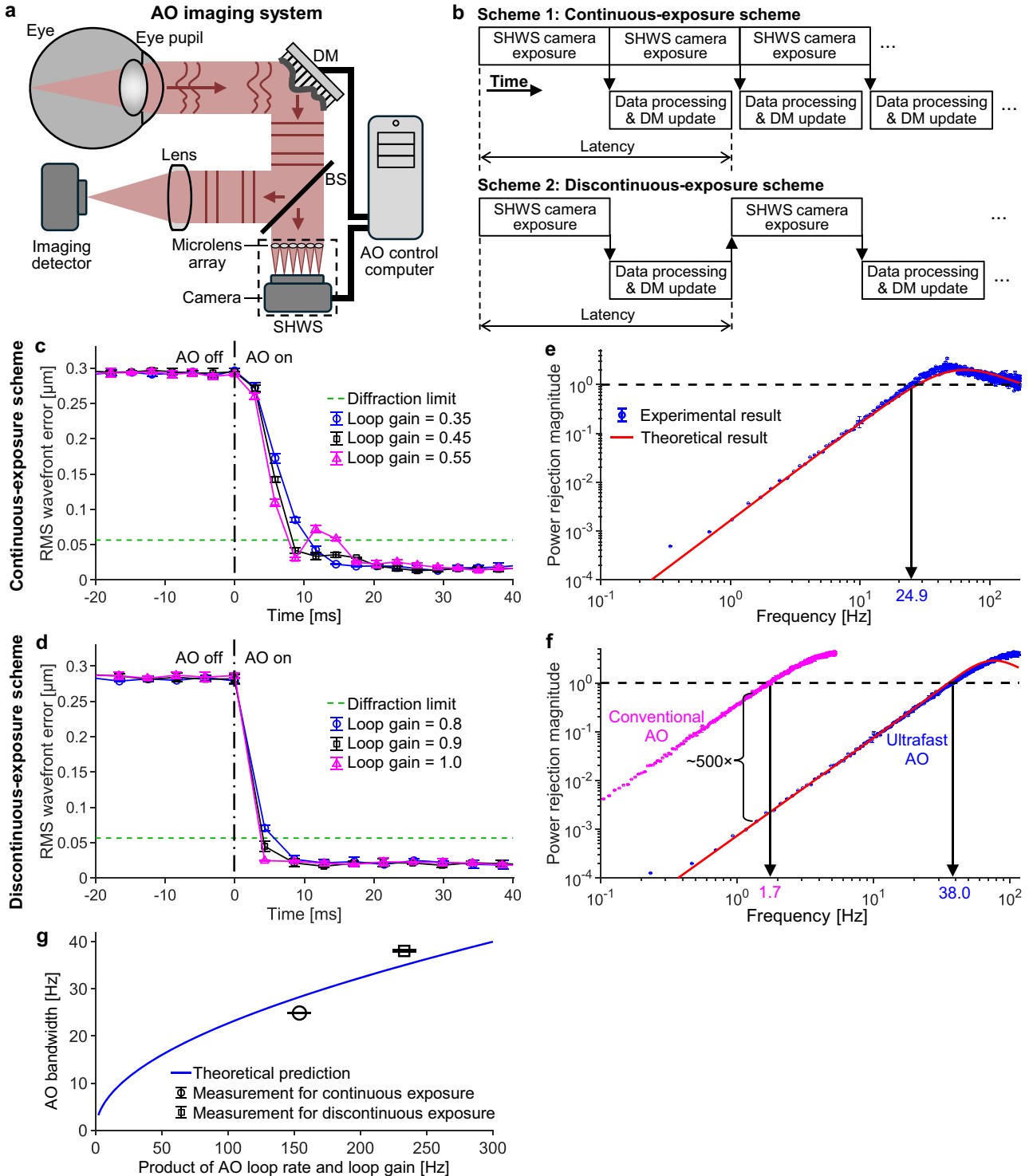

**Fig. 1 | Comparison of two fundamental AO operational schemes and their temporal performances. a** Illustration of a closed-loop AO system. Illumination path, relay optics, and scanners are omitted. Eye pupil is conjugate with DM and SHWS microlens array. BS beamsplitter; DM deformable mirror; SHWS Shack-Hartmann wavefront sensor. **b** Timing diagrams for the two AO operational schemes. RMS wavefront error over time before and after AO was turned on (at time 0) to correct a model eye aberration for continuous exposure (**c**) and discontinuous exposure (**d**), respectively. Different loop gains were tested to determine the optimal value. The power rejection curve when the AO ran at the optimal loop gain for the continuous- (**e**) and discontinuous-exposure (**f**) scheme, respectively. In (**f**), the power rejection curve of conventional AO is also shown for comparison with that of ultrafast AO. The theoretical curves shown in (**e**) and (**f**) are calculated using Eq. (1). Error bars and their centers in (**c**)−(**f**) represent the standard deviation and mean of three measurements, respectively. Most error bars are smaller than the markers. **g** AO bandwidth is a monotonically increasing function of the rate-gain product, as shown by the blue curve calculated from Eq. (2). Markers denote experimental results. Error bars and their centers represent the standard deviation and mean of three measurements, respectively.

We then measured the power rejection curve and AO bandwidth with the system running at the optimal loop gain for each operational scheme (see "Methods"). Figure 1e, f shows that the measured power rejection curves match closely with theoretical predictions (calculated by Eq. (1) shown later). AO bandwidths were measured to be 24.9 Hz and 38.0 Hz for the continuous- and discontinuous-exposure scheme, respectively, close to their theoretically predicted bandwidths of 28.2 Hz and 35.0 Hz (calculated by Eq. (2) shown later). Notably, we find that the discontinuous-exposure scheme experimentally achieves a 53% larger AO bandwidth than the continuous-exposure scheme, even though the AO loop rate of discontinuous exposure is 32% lower.

We note that the product of the AO loop rate and loop gain under discontinuous-exposure scheme (=233 Hz × 1.0 = 233.0 Hz) is also larger than that under continuous-exposure scheme (=342 Hz × 0.45 = 153.9 Hz). This result suggests that this product might explain the larger AO bandwidth achieved by the discontinuous-exposure scheme and may therefore be fundamental to AO system temporal performance. As we were unable to find this relationship in the literature, we theoretically tested the correctness of this supposition, beginning with the derivation of the expression for the power rejection curve $|H_{\text{reject}}(s)|^2$ that is presented in Supplementary Note 2. The resulting expression, Eq. (1), applies to both continuous- and discontinuous-exposure schemes and reveals that the power rejection curve depends on five system parameters ($T_{\text{integration}}$, $T_{\text{DM}}$, $T_{\text{delay}}$, loop rate, and loop gain):

$$|H_{\text{reject}}(s)|^2 = \frac{1}{\left|1 + \frac{1-\exp(-sT_{\text{integration}})}{sT_{\text{integration}}} \frac{\exp(-sT_{\text{delay}})}{s} \frac{1}{1+T_{\text{DM}}s/(2\pi i)} \times (\text{loop rate} \times \text{loop gain})\right|^2},$$

(1)

where $s = i2\pi f$, $i^2 = -1$, $f$ is the temporal frequency, $T_{\text{integration}}$ (= 0.126 ms) is the integration time of the SHWS camera, $T_{\text{DM}}$ (=0.55 ms) is the time constant for DM actuation (we modeled the DM as a low-pass filter and used the DM response curve from the manufacturer, see Supplementary Note 2), and $T_{\text{delay}}$ (=2.42 ms) is the combined duration of camera readout, camera-to-computer data transfer, and data processing (see Supplementary Fig. 1 for a timing diagram). $T_{\text{integration}}$, $T_{\text{DM}}$, and $T_{\text{delay}}$ are the same for both the continuous- and discontinuous-exposure schemes. Therefore, the differences in the power rejection curve and AO bandwidth between the two schemes must be caused by the difference in the product of loop rate and loop gain (the rate-gain product).

We further confirm this assertion and quantify the relationship between AO bandwidth $f_c$ and rate-gain product by setting $|H_{\text{reject}}(s)|^2$ in Eq. (1) equal to 1 (following the definition of AO bandwidth). This derivation is detailed in Supplementary Note 3 and results in the expression:

$$T_{\text{integration}}(2\pi f_c)^2(1+T_{\text{DM}}f_c) \times \sin\left[\pi f_c\left(2T_{\text{delay}} + T_{\text{integration}}\right)\right]$$
$$-(\text{loop rate} \times \text{loop gain}) \times \sin\left(\pi f_c T_{\text{integration}}\right) = 0.$$

(2)

Equation (2) is unfortunately a transcendental equation with no closed-form solution for $f_c$. We solve it numerically and plot $f_c$ as a function of the rate-gain product in Fig. 1g, which shows that AO bandwidth is a monotonically increasing function of the rate-gain product. As seen from Eqs. (1) and (2) and Fig. 1g, the AO loop rate and loop gain are equally important in determining the temporal performance of AO, and optimizing their product is necessary to improve system performance.

Because the discontinuous-exposure scheme achieves a larger rate-gain product than the continuous-exposure scheme does, it achieves a larger AO bandwidth. Hence, we used the discontinuous-exposure scheme for in vivo human retinal imaging.

## DM actuation degrades AO performance and is mitigated by using an optimal exposure delay under discontinuous exposure but not continuous exposure

The short wavefront sensor integration time used in ultrafast ophthalmic AO reduces latency, but it can also increase the system's sensitivity to potential DM actuation effects. If the wavefront sensor camera exposes during DM actuation and if the integration time is shorter than or comparable to the DM actuation time, the wavefront sensor will measure an aberration different from that when the DM fully settles. Hence, DM actuation can corrupt the wavefront sensor measurement and degrade AO performance. The discontinuous-exposure scheme allows us to control the wavefront sensor exposure start time relative to when DM actuation begins, unlike continuous-exposure scheme that unavoidably exposes during DM actuation; we can vary this time by delaying exposure to study the DM actuation effect and minimize it with an appropriate exposure delay. In Supplementary Note 4, we show that DM actuation degrades AO performance and present a method to find the optimal exposure delay (0.3 ms) that minimizes AO overshoot caused by DM actuation while maximizing AO bandwidth. The results in Fig. 1d, f were achieved using this optimal delay.

## Ultrafast AO outperforms conventional AO by an order of magnitude or more on key speed metrics

The most important quantity for characterizing the temporal performance of an AO system is the power rejection curve[1,18,35,55,60]. Figure 1f shows theoretical and experimental power rejection curves for mimicked conventional AO and our ultrafast AO. We mimicked a conventional ophthalmic AO system by slowing our ultrafast ophthalmic AO system, specifically using a wavefront sensor integration time of 45 ms, an AO loop rate of 10 Hz, and an AO loop gain of 1. These parameters are based on the average parameters of the two most widely used ophthalmic AO systems in academia[32,38] and in industry[33]. The former uses a wavefront sensor integration time of 60 ms, an AO loop rate of 8.7 Hz, a loop gain of 0.9−1 and has a theoretical maximum AO bandwidth of 1.2 Hz with a loop gain of 1; the latter uses a sensor integration time of 30 ms, AO loop rate of 10−12 Hz, loop gain of 0.3−1, and has a theoretical maximum AO bandwidth of 1.6 Hz when the loop rate is 12 Hz and the loop gain is 1. Both systems use the discontinuous-exposure scheme, but this information has not been described in the literature and was obtained from personal communications.

Figure 1f shows that the experimentally-achieved bandwidths of mimicked conventional AO and ultrafast AO are 1.7 Hz and 38.0 Hz, respectively, close to their theoretical predictions of 1.4 Hz and 35.0 Hz. In addition to achieving a much larger AO bandwidth than conventional AO, ultrafast AO also reduces aberration power density 500× more at each temporal frequency at which conventional AO corrects aberrations.

Figure 2 depicts typical ultrafast ophthalmic AO performance on three healthy young subjects without cycloplegia. The figure also includes simultaneously-acquired AO optical coherence tomography (AO-OCT) images of the subjects' cone photoreceptors (see Methods for a description of our AO-OCT system). Both RMS wavefront error (Fig. 2a−c) and the effect on retinal image quality (Fig. 2 d−f) are shown in the time immediately before and after the AO loop is closed; the time axis of the plot has been scaled to match the slow-scan acquisition rate of the image. After AO activation, RMS wavefront error dropped from -0.29 to 0.45 µm to below the diffraction limit (= $\lambda/14$ = 0.056 µm) within 10 ms (Fig. 2a−c), more than 20× faster than the fastest ophthalmic AO system reported in the literature[53]. Substantiating the wavefront sensor measurements, sharp AO-OCT imaging of the cone mosaic occurred within a few milliseconds following AO activation (Fig. 2d−f).

Table 1 (columns 2 and 3) compares key speed-related performance parameters of our ultrafast ophthalmic AO system with corresponding values for conventional ophthalmic AO. Ultrafast AO

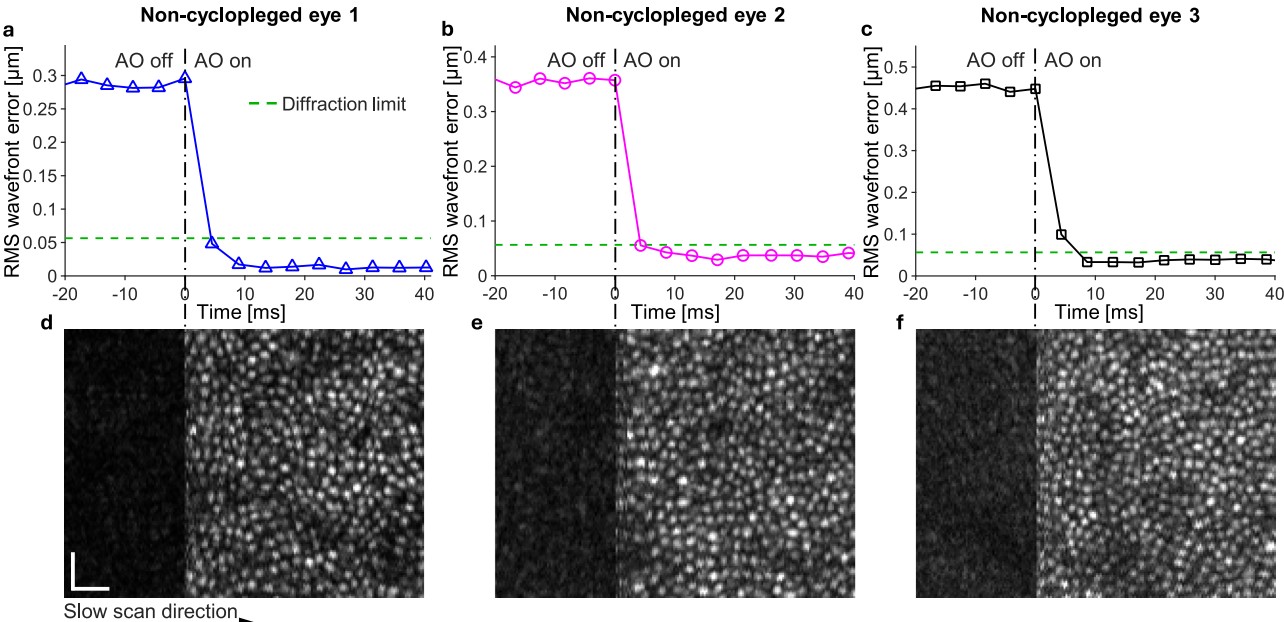

**Fig. 2 | Time for ultrafast ophthalmic AO performance to reach diffraction limit in non-cyclopleged eyes. a–c** RMS wavefront error converges to diffraction limit within 10 ms in non-cyclopleged eyes of three healthy young subjects (aged 28–33 years). **d–f** The corresponding AO-OCT en face images of the cone mosaic are shown during AO activation. **a–c** Plots and **d–f** images share the same time axis; time values of the data points correspond to when control voltages were sent to the DM. Scale bars = 20 μm.

outperforms conventional AO by roughly an order of magnitude or more on all parameters, with key ones including latency, time to reach diffraction limit, and AO bandwidth.

### Noise level of our ultrafast ophthalmic AO system has a negligible effect on AO performance

Our ultrafast AO uses an integration time (0.126 ms) that is 100–500× shorter than other ophthalmic AO systems to reduce AO latency, resulting in a dimmer wavefront sensor image with lower SNR. Despite this, we show that the RMS wavefront error due to this lower SNR is still much below the diffraction limit, implying that photon and camera noise have a negligible effect on AO performance.

Even with the short integration time used by our wavefront sensor, Figure 3a shows the focal spots of our SHWS image to be of high quality. SHWS images from our 24 subjects (subject information is shown in Supplementary Table 1) were determined to contain $1,962 \pm 673$ (Mean ± SD) photons per lenslet per 0.126-ms frame. By comparison, astronomers typically use 100 photons per lenslet per frame for AO[60,65], 20× fewer. By using Eqs. (M1) and (M2) in Supplementary Note 5, the wavefront sensing errors due to photon shot noise $\sigma_{\varphi,\text{photon}}$ and camera readout noise $\sigma_{\varphi,\text{readout}}$ are $0.100 \pm 0.052$ rad and $0.010 \pm 0.003$ rad, respectively, with the photon shot noise dominating. Total wavefront sensing error $\sigma_{\varphi,\text{total}}$ is $0.100 \pm 0.052$ rad.

To study the impact of this wavefront sensing error on the residual wavefront error seen by the retinal imaging camera, we need to know how wavefront sensing error is propagated through the AO loop. This error propagation is quantified by the noise transfer function. Figure 3b shows noise transfer functions for conventional and ultrafast AO using the previously described experimental parameters and calculated using Equation (M3) in Supplementary Note 5. Although ultrafast AO uses a loop gain of 1, the noise transfer function shows almost no amplification of wavefront sensor noise through the AO loop and exhibits less overshoot than that of conventional AO. Note that our ultrafast AO system can use a loop gain of 1 with almost no noise amplification because we use the discontinuous-exposure scheme. If we were to use the continuous-exposure scheme with a

two-frame delay (i.e., $T_{\text{integration}} = T_{\text{delay}} = 1/342$ ms) commonly used in astronomy and by some retinal imaging groups[55,56,60], the AO system with a loop gain of 1 would significantly amplify the wavefront sensing noise as indicated by the red curve in Fig. 3b.

Considering both the wavefront sensing error and noise transfer function, we use Equation (M4) in Supplementary Note 5 to estimate the RMS wavefront error due to wavefront sensing noise to be $12.6 \pm 6.6$ nm (Mean ± SD, 24 subjects). This is much smaller than the diffraction limit ($\lambda/14 = 56$ nm), and as a result, we can conclude that noise in the SHWS image has essentially no impact on AO performance.

### Characterizing ocular aberration dynamics up to 171 Hz in clinically-relevant scenarios and demonstrating the need for faster AO in these scenarios

Because the temporal spectrum of ocular aberrations in clinically-relevant scenarios is poorly understood, we used the wavefront sensor of our ultrafast ophthalmic AO system to measure ocular aberration dynamics at high temporal resolution. We conducted 35 subject measurements across 24 different subjects, covering six targeted scenarios, which we refer to as "clinically relevant," along with one control scenario.

The six clinically-relevant scenarios we investigated were:

1. Normal eye with artificial tears. Eye drops are commonly administered to patients suffering from dry eye, a condition caused by tear deficiency or excessive tear evaporation and often secondary to other medical issues. These drops help maintain moisture on the corneal surface and are often used during retinal imaging examinations. However, the artificial tears that provide the greatest comfort are generally the most viscous, resulting in blurry vision and blurry retinal images, especially after a blink.
2. Normal eye without cycloplegia. A significant number of patients cannot use cycloplegic drops. This includes those with closed-angle glaucoma, an anterior chamber intraocular lens, or an occludable angle, as these conditions can impair aqueous humor outflow and pose serious risks. Also, cycloplegic drops are avoided in pregnant or nursing women due to potential health

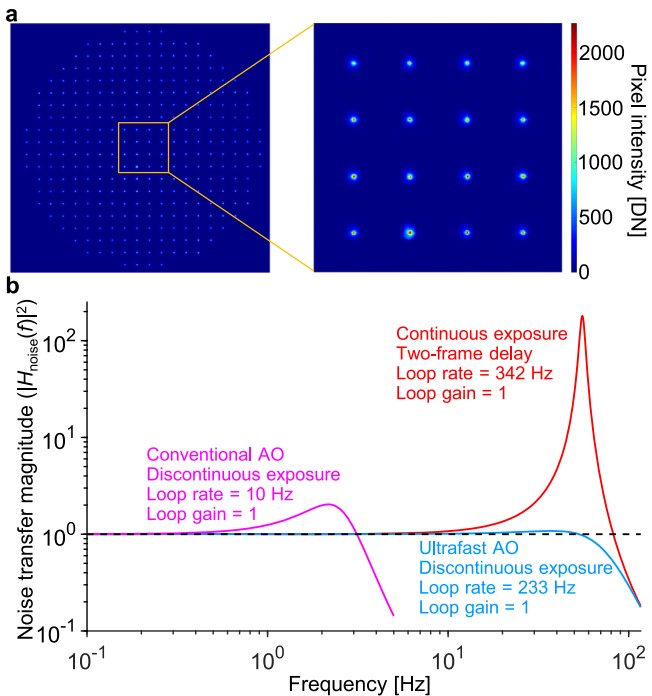

**Fig. 3 | Noise analysis of ultrafast ophthalmic AO. a** Raw SHWS image acquired from a human eye after subtracting a dark image acquired with no input light. Even with a short wavefront sensor integration time of 0.126 ms, the SHWS image acquired with AO off shows well-defined focal spots. DN, digital number. **b** Squared magnitude of the noise transfer functions of conventional and ultrafast AO. The sampling rates (AO loop rates) were 10 Hz and 233 Hz for conventional and ultrafast AO, respectively, using the discontinuous-exposure scheme. The sampling rate was 342 Hz using the continuous-exposure scheme.

concerns for the fetus or infant. Many patients also find these drops uncomfortable and disruptive to their daily activities, as their effect can last for several hours.

3. Normal eye with sequential fixation. AO imaging is limited to the eye's isoplanatic patch size (-1°)[66], which is much smaller than images acquired with clinical ophthalmoscopes (>20°). Montaging − the integration of multiple images to expand the instrument's effective field of view − is a common method to address the eye's small isoplanatic patch size. However, this technique requires the subject (or instrument) to rapidly change fixation, which can introduce additional aberrations. In this scenario, we mimic the image acquisition process used in montaging.

4. Keratoconic eye with contact lens. Keratoconus is a progressive eye disease in which the cornea thins and takes on a conical shape, causing vision loss. It is a leading cause of corneal transplantation[50]. To improve vision, a hard contact lens is commonly used to help neutralize the aberrations of the irregularly-shaped cornea. However, the contact lens moves about on the eye, especially after a blink, causing extratemporal aberrations in addition to those caused by eye movement.

5. Myopic eye with contact lens. High myopia (≥6 diopters in magnitude) affects a significant portion of the global population (163 million people as of 2020) and is projected to affect 1 billion people (or 10% of the global population) by 2050[67]. Unfortunately, standard methods for correcting refractive errors, such as using trial lenses or a Badal system in front of the eye, cannot be easily integrated into mirror-based AO imaging systems[44,68]. Existing DMs still lack the dynamic range needed to correct high myopia. Contact lens correction thus offers a potentially attractive and clinically viable alternative for imaging these

subjects by helping to conserve the dynamic range of the DM for correction of higher-order aberrations.

6. Nystagmic eye. Nystagmus is a condition characterized by involuntarily, rhythmic eye movements and may be associated with serious health issues, especially those affecting the brain. Due to the constant motion, imaging nystagmic eyes with AO retinal imaging systems is notoriously challenging[44,69].

To measure the ocular aberration dynamics in these scenarios, we operated the wavefront sensor at 342 Hz under its internal-trigger mode, and set the integration time to 1/342 Hz = 2.9 ms to minimize potential aliasing. For the noise performance quantification presented below, we used neutral density filters to equate SHWS photon count during the 2.9 ms exposure to that in the AO-OCT imaging experiments with 0.126 ms exposure. Figure 4a shows the power spectra of the ocular aberrations we measured, plotted as the mean of five subjects with standard error for each scenario (individual subject data is shown in Supplementary Fig. 2). For comparison, mean power spectra of normal healthy eyes under the typical AO imaging scenario (i.e., subjects are cyclopleged, fixating at a single location, and blink prior to a 5-s data acquisition period) are also shown. In all conditions, the power spectral density (PSD) curves decrease with increasing temporal frequency (following a power law indicating less power at higher frequencies) until they reach a plateau determined by the noise floor. Note however, the curve for the nystagmic eyes deviates somewhat from this trend, showing increased power at 3−8 Hz that is generated by the different oscillatory motions of the different nystagmic subjects (see individual subject curves in Supplementary Fig. 2b). The six clinically-relevant scenarios exhibit up to two orders of magnitude higher aberration PSD than the normal group at all temporal frequencies before reaching the noise floor.

To aid interpretation, the diffraction limit threshold for conventional AO (the black dashed line in Fig. 4a) indicates the PSD spectrum of an input aberration that conventional AO can barely correct to achieve diffraction-limited performance (i.e., RMS wavefront error averaged over the measurement duration is equal to $\lambda/14$). As will be shown in Fig. 4c, conventional AO is too slow to achieve diffraction limit when an ocular aberration PSD spectrum is above the diffraction-limit threshold. Thus, conventional AO is unable to achieve diffraction-limited performance for any of the clinically-relevant scenarios shown in the figure. By contrast, the diffraction limit threshold for ultrafast AO (shown as the black dotted line in Fig. 4a) falls above all the ocular aberration power spectra (except for part of the Nystagmic trace), indicating that ultrafast AO can achieve diffraction-limited performance for all the clinically-relevant scenarios (at least when quantified by mean PSD value).

As a side note relevant to the previous subsection, the noise floors represented by the high-frequency plateaus in the PSD spectra in Fig. 4a allow us to more directly characterize the noise-induced RMS wavefront error by $\sigma_W = \sqrt{PSD_{noise} \times BW}$, where $PSD_{noise}$ is the PSD amplitude of the noise floor and BW = 342/2 Hz is the spectral bandwidth due to sampling[27,70]. The results for the six scenarios and control shown in Fig. 4a are presented in Supplementary Fig. 3, with a Mean ± SD of 23.3 nm ± 8.1 nm (24 subjects), smaller than the diffraction limit (56 nm). This provides additional confirmatory evidence that the noise effect on wavefront sensing performance is inconsequential, even though we use a short wavefront sensor integration time to reduce latency.

Figure 4b shows simulated power rejection curves for AO systems with different loop speeds. In the simulation, we assume that for each AO system, the wavefront sensor integration time takes up half of the loop period, with the other half used by the combined data readout, transfer and processing times[53,55,60]. Discontinuous exposure with an AO loop gain of 1 is used in the simulation. It is clear from Fig. 4b that a

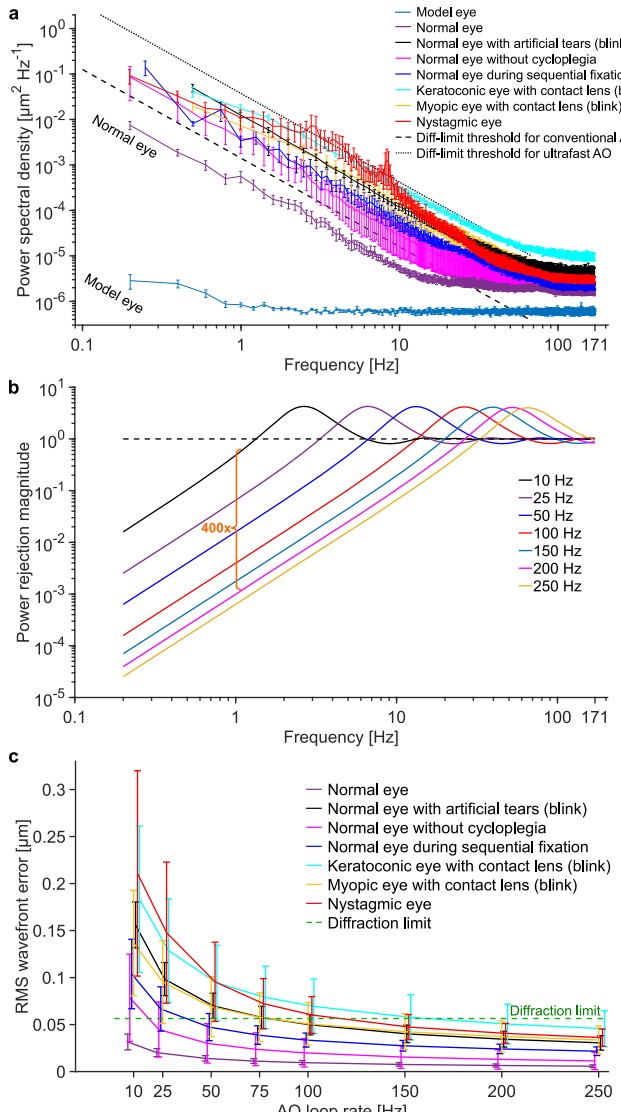

**Fig. 4 | Power spectra of ocular aberrations and required AO speed for correcting the aberrations are significantly higher for the clinically relevant scenarios compared to the normal group. a** Measured power spectra of ocular aberrations are shown for normal and six clinically-relevant scenarios. Mean of five subjects for each scenario is shown and error bars represent the standard error. Individual subject data is shown in Supplementary Fig. 2. All eyes were dilated and cyclopleged with 1% Tropicamide, except the one labeled otherwise. The artificial tears were Refresh Optive® gel drops; they were the most comfortable but also the most viscous to the subject out of the three types we tested (described later). All power spectra measurements were based on 5-second long videos, except for 1) the blink scenarios, for which we analyzed the 2-second long data acquired after the eye reopened from blinks and 2) the sequential fixation scenario, for which we analyzed the 4-second long video. **b** Simulated power rejection curves for AO systems with different loop rates. Higher loop rates correct higher temporal frequencies and more effectively reduce lower temporal frequencies. See main text for AO system details. **c** Predicted AO performance in terms of RMS wavefront error as a function of AO loop rate for the different scenarios, using the different power rejection curves in (**b**). Data points from different scenarios were shifted horizontally to improve visibility. Error bars and their centers represent the standard deviation and mean of five subjects.

faster AO system not only corrects aberrations of higher temporal frequencies (i.e., increases the AO bandwidth), but also corrects aberrations orders-of-magnitude more effectively at lower frequencies (i.e., has lower power rejection magnitudes). The latter is very helpful because ocular aberrations have more power at lower frequencies

regardless of scenario as shown in Fig. 4a and by others[26,27]. Compared with conventional AO running at 10 Hz, an AO system running at 200 Hz reduces the input aberration power density by ~400× more at all frequencies where conventional AO corrects aberrations. This results in lower residual wavefront aberrations, as will be shown in Fig. 4c.

To predict how fast an AO system would need to run in order to handle each of the ocular aberration dynamics characterized by the power spectra $PSD_{eye}(f)$ shown in Fig. 4a, we calculated RMS wavefront error (i.e., the temporal error $\sigma_{temporal}$) after correction by AO with different speeds. The AO systems with different speeds are characterized by the power rejection curves in Fig. 4b and we used Equation (M5) in Supplementary Note 6 to calculate the wavefront errors. The results for the mean performance and standard deviation in performance across subjects are shown in Fig. 4c. For the normal group, conventional AO running at 10 Hz with discontinuous exposure is fast enough to achieve diffraction-limited performance (averaged over a measurement window of 5 s) for all of the subjects. In contrast, for the six clinically-relevant scenarios that exhibited higher aberration power spectra in Fig. 4a, both the mean and standard deviation of their residual RMS wavefront error are larger at any given AO loop rate than for the control group. Based on the mean RMS performance, the same 10-Hz conventional AO system is too slow for any of the six scenarios. The following loop rates are predicted to achieve diffraction-limited performance: 160 Hz for keratoconic eye with contact lens (blink), 115 Hz for nystagmic eye, 80 Hz for normal eye with viscous artificial tears (blink), 80 Hz for myopic eye with contact lens (blink), 40 Hz for normal eye during sequential fixation, and 20 Hz for normal eye without cycloplegia. Note that these speed requirements are predicted for the mean subject, and therefore even higher speeds would be needed to capture a larger proportion of the population. With only five subjects in each scenario, it is difficult to generalize our findings to a larger population. Nevertheless, we can roughly estimate the loop rates required for 95% of the population: >250 Hz for keratoconic eye with contact lens (blink), ~250 Hz for myopic eye with contact lens (blink), ~200 Hz for nystagmic eye, 150 Hz for normal eye with viscous artificial tears (blink), ~75 Hz for normal eye without cycloplegia, and 75 Hz for normal eye during sequential fixation. These loop rates were obtained using 95% one-tailed confidence interval of 1.65 times the 5-subject standard deviation in each condition. Note that the AO loop rates discussed here correspond to AO systems using the discontinuous-exposure scheme with a loop gain of 1. If an AO system uses the continuous-exposure scheme, which typically uses a loop gain of 0.3 − 0.5 to maintain stability, the required AO loop rates are even higher. These results demonstrate the necessity of faster AO.

The following four sub-sections demonstrate the benefit of increasing AO bandwidth and power rejection performance. The subsections compare the performance of ultrafast ophthalmic AO to conventional ophthalmic AO for the six clinically-relevant scenarios. For brevity, figures show the result from a representative subject for each scenario, defined as the subject with the median aberration power spectrum of the five subjects measured in each scenario (see Supplementary Fig. 2).

## Ultrafast ophthalmic AO enables a more stable focus and fine focus control in non-cycloplegic eyes

The focus of the eye is known to fluctuate during steady-state accommodation[27,71]. Such fluctuation makes it difficult to image a specific retinal layer accurately and stably over time, especially with AO ophthalmoscopy which has a much smaller depth of focus than non-AO ophthalmoscopy. We find that ultrafast AO greatly improves focus stability in non-cycloplegic eyes, even when subjects fixated on a target at or near their far points. Figure 5a shows consecutive retinal

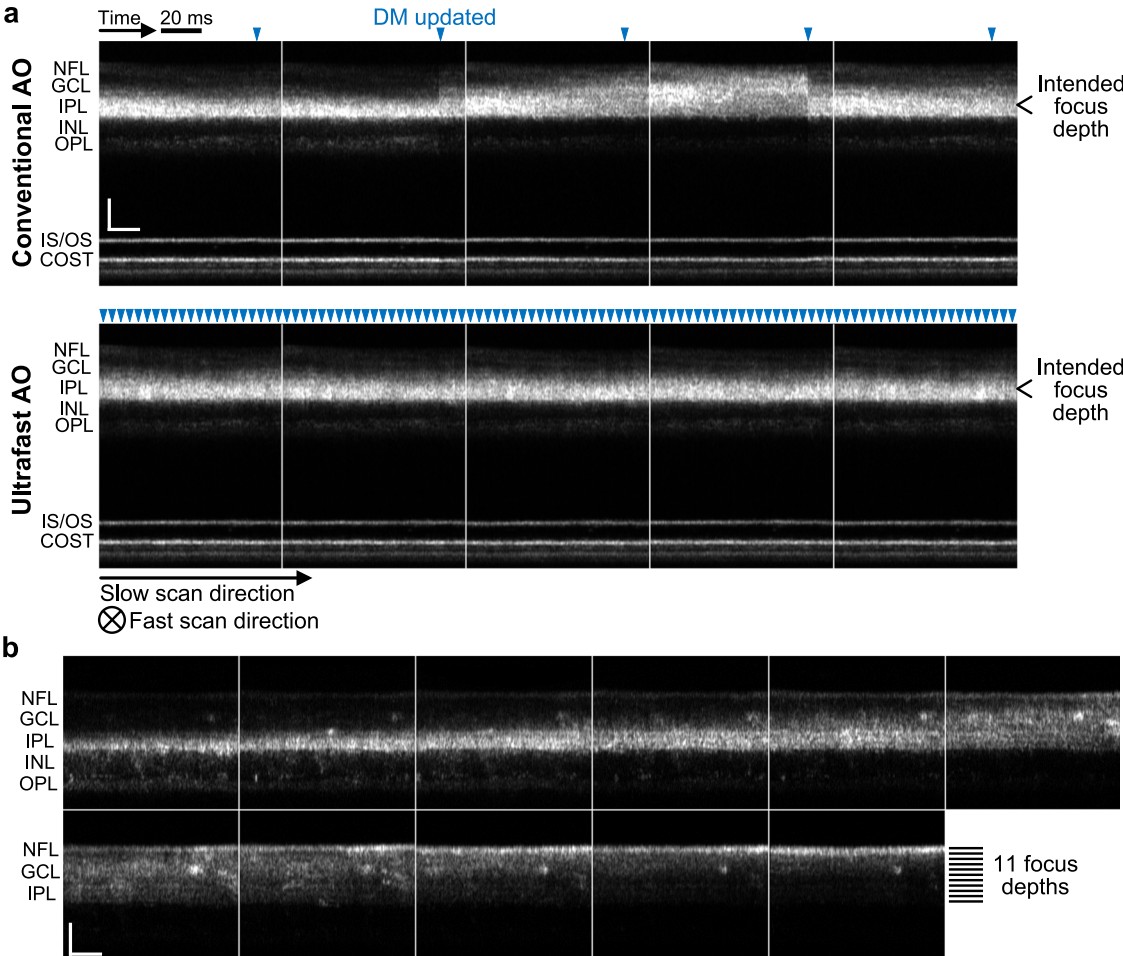

**Fig. 5 | Ultrafast ophthalmic AO provides better focus stability and finer focus control in non-cyclopleged eyes. a** Consecutive AO-OCT retinal images acquired from a healthy 25-year-old male with conventional AO and ultrafast AO, respectively. See Supplementary Movie 1 for the full video. Each image is a projection of a volume along the fast-scan axis, and axial displacement of the retina between fast B-scans was corrected by aligning the photoreceptor layers. Blue arrowheads denote the time points when the DM actuators were updated. Note that AO updates within the OCT volume instead of at the beginning or end, thus the jump of system focus for conventional AO (indication of insufficient AO bandwidth) is more clearly seen in each image. COST cone outer segment tip, GCL ganglion cell layer, INL inner nuclear layer, IPL inner plexiform layer, IS/OS inner segment/outer segment junction, NFL nerve fiber layer, OPL outer plexiform layer. Scale bars = 50 µm. **b** Ultrafast AO enables fine focus control in the inner retina of a healthy 30-year-old male. The AO-OCT beam was focused at 11 depths sequentially from IPL to NFL with a step size of 0.02 D (≈7.4 µm). Each image is a projection of a cropped AO-OCT volume (the central 141 pixels of each fast B-scan) along the fast-scan axis. Scale bars = 50 µm.

AO-OCT images acquired in the subject with the median aberration power spectrum, using conventional AO and ultrafast AO with the focus set at the inner plexiform layer (IPL). Each image is a projection of a volume along the fast-scan axis. Optical focus depth clearly fluctuates over time when using conventional AO (Fig. 5a and Supplementary Movie 1), varying between the inner nuclear layer (INL) and the nerve fiber layer (NFL). With ultrafast AO, by contrast, the focus depth is an order of magnitude more stable and limited to the IPL (Fig. 5a and Supplementary Movie 1).

The stable focus provided by ultrafast ophthalmic AO enables fine focus control within the 200–300 µm thickness of the retina. Figure 5b and Supplementary Movie 2 show cross-sectional AO-OCT images of the inner retina acquired while we sequentially focused light at 11 different depths from IPL to NFL with a step size of 0.02 D (≈7.4 µm). We observe differences between neighboring images even though the axial focal shift between them is only 7.4 µm. This capability is useful for targeting the cellular composition of different retinal layers and sublayers that are affected by different diseases, such as the ganglion cell and nerve fiber layers in glaucoma, the vascular plexuses in diabetic retinopathy, and the photoreceptor components (inner segment,

outer segment, soma, axons) in age-related macular degeneration and retinitis pigmentosa.

## Ultrafast ophthalmic AO converges an order of magnitude faster after a blink, improving imaging throughput in normal, myopic and keratoconic eyes

We compare the image quality and performance of conventional and ultrafast AO immediately after eye blinks and in the presence of a contact lens (used in myopic or keratoconic eyes) or viscous artificial tears (used in normal eyes). Figure 6a shows consecutive AO-OCT en face images of a non-cyclopleged healthy eye just after a blink, using conventional (top) and ultrafast AO (bottom). Image quality of the first two frames is visibly poorer with conventional AO due to its inability to correct the additional aberrations that occur after a blink. By contrast, ultrafast AO achieves high image quality immediately after the blink. The RMS wavefront error as a function of time averaged over 12 trials is shown in Fig. 6b. After the eye reopened, ultrafast AO reached the diffraction limit in 14 ms, 14× faster than conventional AO (197 ms). The ability to capture sharp images immediately after a blink enables near-continuous data acquisition between blinks for simpler and

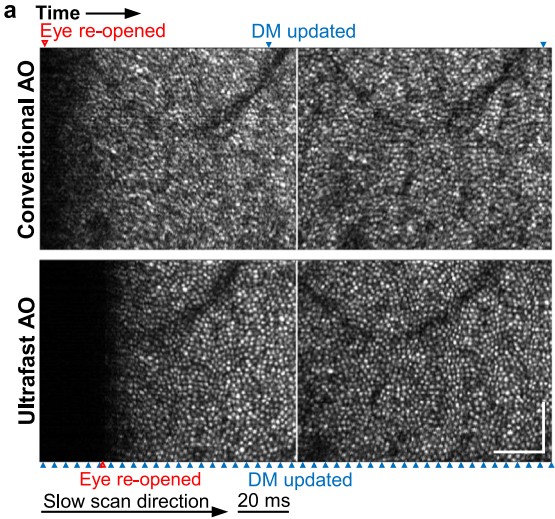

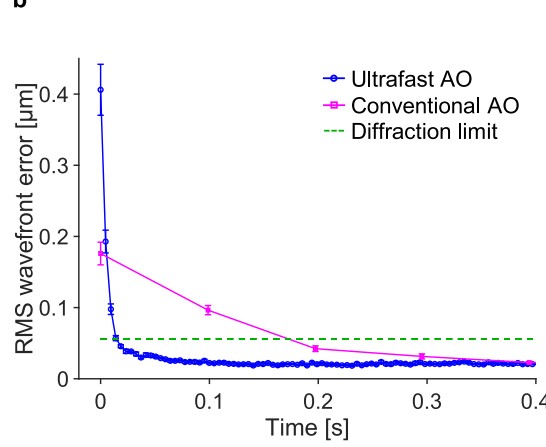

**Fig. 6 | Ultrafast ophthalmic AO converges faster following an eye blink by a healthy subject. a** Consecutive AO-OCT en face images of the cone mosaic captured at 10 Hz of a non-cyclopleged eye that has re-opened from a blink. Blue and red arrowheads denote the time points when the DM was updated and the eye re-opened, respectively. Scale bars = 60 μm. **b** RMS wavefront error over time when the eye re-opened from a blink (0 ms), averaged over 12 measurements that were made on the same eye. Error bars represent the standard errors.

higher-throughput imaging in the clinic and in the laboratory. This could be especially useful in older subjects whose tear films are unstable or in those suffering from dry eye, which requires frequent blinking. In such cases, ultrafast AO can improve image quality immediately after each blink, continuing until the tear film breaks, with the benefits diminishing thereafter until the next blink.

The ultrafast-AO advantage is maintained in the presence of a contact lens. Contact lenses are notorious for moving about on the cornea, especially after a blink. To assess the ability of our AO system to correct the resulting aberrations, we imaged five high myopes while they wore contact lenses during AO-OCT imaging. Figure 7a shows consecutive AO-OCT en face images acquired immediately after a blink, using conventional AO (top) and ultrafast AO (bottom), for a cyclopleged 27-year-old female subject with a large refractive error of −6.5 D (subject with median aberration power spectrum). Image quality of the first three frames after a blink is substantially poorer with conventional AO, while ultrafast AO achieves high image quality quickly after the blink. The RMS wavefront error as a function of timeaveraged over 12 trials is shown in Fig. 7b. Ultrafast AO reached diffraction limit in ~27 ms, 11× faster than conventional AO (~300 ms).

We also imaged five keratoconic subjects while they wore contact lenses during imaging. Supplementary Movie 3 shows the apparent movement of the contact lens on the eye after a blink. Figure 7c shows six consecutive AO-OCT en face images taken just after a blink, using conventional AO (top) and ultrafast AO (bottom), for the subject with the median aberration power spectrum. Image quality of the first five frames after a blink is substantially poorer with conventional AO, whose bandwidth is insufficient as indicated by the abrupt jumps in image quality with each DM update and the overshoot (bump) in the RMS wavefront error shown in Fig. 7d. The overshoot suggests that the loop gain of 1 is too high with conventional AO for this scenario. In contrast, we did not observe similar performance issues with ultrafast AO, which achieved high image quality quickly after the blink.

The ultrafast-AO advantage is also maintained in the presence of viscous artificial tears. To test the possible benefits of ultrafast AO, we administered the following three artificial tears sequentially to the same subject (with rinsing by saline solution in between): 1. Refresh RELIEVA® (active ingredients (AI): carboxymethylcellulose sodium (CS) 0.5% + glycerin 0.9%); 2. Refresh CELLUVISC® (AI: CS 1%); and 3. Refresh Optive® gel drops (AI: CS 1% + glycerin 0.9%). Out of the three artificial tears, the subject reported that the third one was the most comfortable, but also the most viscous. We found that these high-viscosity artificial tears induced large temporal aberrations after blinks (see power spectrum in Fig. 4a). Figure 7f shows six consecutive AO-OCT en face images taken just after a blink, using conventional AO (top) and ultrafast AO (bottom), for the subject with the median aberration power spectrum. Image quality of the first five frames after a blink is substantially poorer with conventional AO and its sixth frame is still worse than that with ultrafast AO. Jumps in image quality with DM updates are also evident and similar to, but not as pronounced as, those observed with the keratoconic subjects. Figure 7e shows the average over 12 trials of post-blink RMS wavefront error as a function of time in an eye bearing the Refresh Optive® artificial tears. Ultrafast AO reached diffraction limit in 55 ms, 12× faster than conventional AO (~700 ms).

## Ultrafast ophthalmic AO converges an order of magnitude faster during sequential fixation

During recording of a 5-s video, we sequentially imaged four retinal locations separated by a distance (2°) larger than the isoplanatic patch size of the eye (typically ~1° for 6-mm pupils[66]). This task mimicked the clinical scenario of rapidly capturing multiple retinal images for montaging. This was accomplished by asking the subject to sequentially fixate for one second at each of the four corners of a cross on a fixation display conjugated to the far point, rapidly moving fixation after an audible beep sounded (Fig. 8a). RMS wavefront errors for conventional AO and ultrafast AO are shown in Fig. 8b, c for the subjects with the median and largest aberration power spectrum among the five subjects measured. When the subjects changed fixation location, RMS wavefront error exceeded the diffraction limit in both AO cases for both subjects, but wavefront error was typically larger and stayed large much longer with conventional AO. Specifically, ultrafast AO recovered to a stable, sub-diffraction-limited error within 15 ms whereas conventional AO took at least 100 ms (one frame of AO correction) to recover. With conventional AO, we also observed large wavefront error spikes exceeding the diffraction limit during fixation (indicated by * in Fig. 8b, c), while any such spikes observed with ultrafast AO (indicated by the arrowheads) were below the diffraction

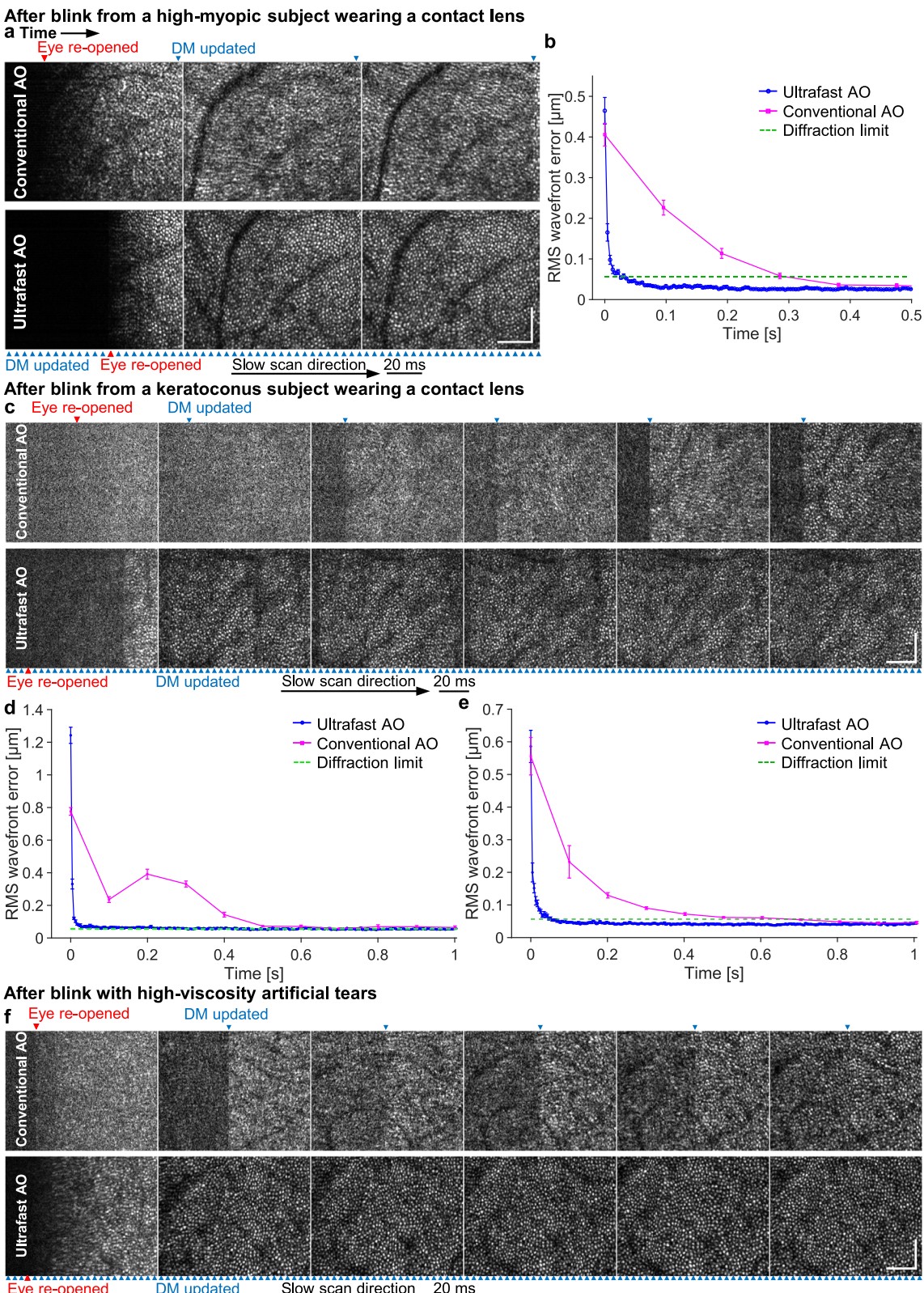

**Fig. 7 | Ultrafast ophthalmic AO converges faster following an eye blink in three additional example scenarios. a** Consecutive AO-OCT en face images of the cone mosaic captured at 10 Hz after eye has re-opened from a blink by a high-myopic subject wearing a − 6.5 D contact lens. Blue and red arrowheads denote the time points when the DM was updated and the eye re-opened, respectively. **b** RMS wavefront error over time when the myopic eye re-opened from a blink (0 ms), averaged over 12 measurements. Error bars represent the standard errors in (**b, d, e**).

**c** Consecutive images of the cone mosaic captured at 10 Hz after eye re-opened from a blink by a keratoconus subject wearing a contact lens. **d** RMS wavefront error over time when the keratoconic eye re-opened from a blink, averaged over 12 measurements. **e** RMS wavefront error over time when an eyebearing viscous artificial tears re-opened from a blink, averaged over 12 measurements. **f** Consecutive images of the cone mosaic captured at 10 Hz after eye has re-opened from a blink by a subject bearing viscous artificial tears. Scale bars = 60 μm in (**a, c, f**).

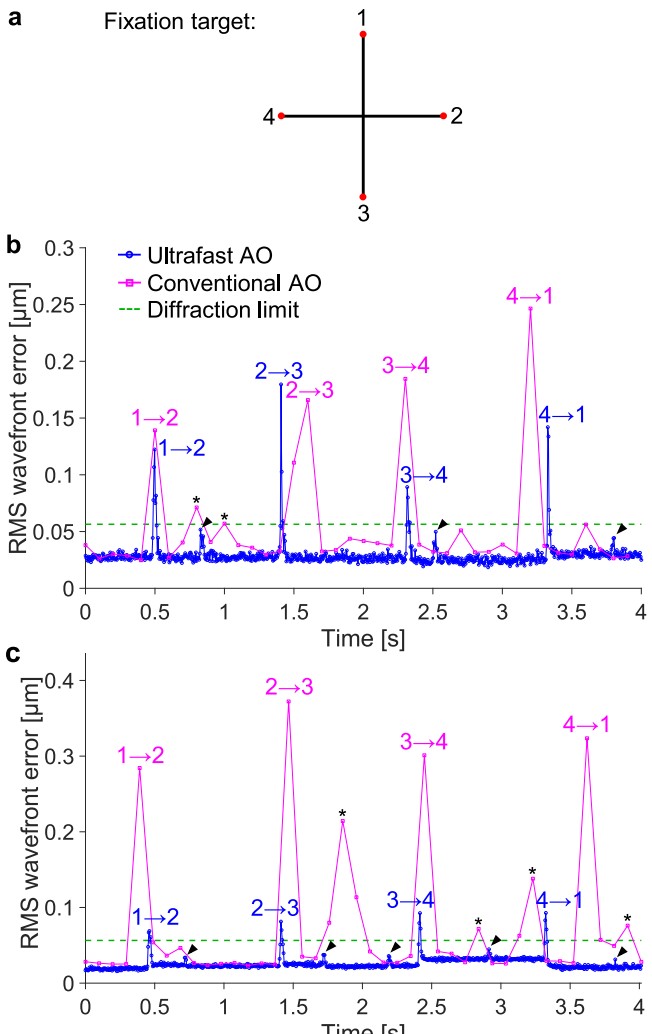

**Fig. 8 | Ultrafast ophthalmic AO converges faster during sequential fixation.**
**a** Subject sequentially fixated for one second at each of the four corners of a cross on a display, moving fixation after an audible beep sounded every second. **b** RMS wavefront error over time for the subject with the median aberration power spectrum among the five subjects measured. Numeric labels, i → j, denote the time immediately after the subject changed his fixation from location i to j. Arrowhead and asterisk symbols denote the wavefront error spikes with ultrafast AO and the wavefront error spikes that are above the diffraction limit with conventional AO, respectively, before the subject changed his fixation to the next location. **c** RMS wavefront error over time for the subject with the largest aberration power spectrum among the five subjects measured. This subject is the oldest of the five and has 2× larger higher-order aberrations as measured with a clinical aberrometer (Pentacam AXL Wave, Oculus).

limit. Because ultrafast AO converges so quickly, we can use the data almost immediately after a change in retinal imaging location. This increases imaging throughput, which enables faster retinal montaging.

## Ultrafast ophthalmic AO improves image quality in the nystagmic eye

Assessing retinal health with AO retinal imaging systems is extremely challenging in eyes that exhibit abnormally fast and large eye movements, such as eyes with nystagmus[44]. We find that ultrafast ophthalmic AO corrects aberrations better and improves image quality of such eyes. Supplementary Movie 4 shows the eye motion of a 26-year-old male subject with congenital nystagmus associated with albinism when

he viewed the fixation target in our AO-OCT system. This subject had the median aberration power spectrum among the five nystagmic subjects measured. The clinical characterization of the nystagmus is moderate amplitude, high frequency, and horizontal jerk to the left. We noticed a reduction in eye motion in our AO-OCT system compared with that under a slit lamp, but the eye motion was still noticeably higher than that of subjects without nystagmus.

We recorded 5-second AO-OCT videos and the corresponding wavefront errors during aberration correction with either conventional or ultrafast AO. The distribution of the RMS wavefront error for both cases averaged over 15 videos is shown in Fig. 9a. The RMS wavefront errors associated with ultrafast AO have a much narrower distribution than those with conventional AO and are shifted towards lower error. The mean wavefront error for ultrafast AO (0.069 μm) is about half of that of conventional AO (0.133 μm), and leads to a 2.3× improvement in Strehl ratio (from 0.32 to 0.74) over conventional AO.

Supplementary Movie 5 shows the cone mosaic videos acquired from this nystagmic eye, after aberration correction with conventional and ultrafast AO. In many volumes acquired with conventional AO, image quality was insufficient to resolve cone cells (e.g., Fig. 9b−e). By contrast, ultrafast AO consistently yielded high image quality (except during microsaccades, which are not correctable with both AO systems), enabling the resolution of many more cells (Fig. 9f−i).

## Discussion
### Ultrafast ophthalmic AO system
We have developed an ultrafast ophthalmic AO system that increases AO bandwidth by ~30× and improves power rejection magnitude by ~500× over conventional ophthalmic AO. As shown in Fig. 1f and Fig. 4b, faster AO not only corrects aberrations of higher frequencies, but also reduces aberrations by orders of magnitude more at lower frequencies where ocular aberrations have the most power (Fig. 4a).

To achieve this superior performance, we targeted specific aspects of the wavefront sensor and control software. We reduced the SHWS latency significantly compared to conventional AO systems by using a wavefront sensor integration time that is 100−500× shorter and a wavefront sensor readout speed that is 3−100× higher. We achieved this speed while maintaining similar number of pixels to preserve spatial fidelity and dynamic range of the wavefront measurement. We also preserved photon-noise-limited imaging without increasing the amount of light entering the eye; this meant reducing the noise level of the camera below that needed in conventional AO. We found that Hamamatsu's ORCA-Lightning camera fit these requirements well but other commercial cameras may too. Even with the Lightning's relatively low quantum efficiency (33%) at 790 nm, the noise level in the wavefront sensor images had negligible effect on AO performance. We imaged 25 subjects across six clinically-relevant scenarios in this study and found the SNR to be sufficient in all cases, that is, the wavefront error due to noise was <λ/14. We have also used the ultrafast AO system to image over 20 subjects with various retinal diseases and eye conditions, including retinitis pigmentosa, age-related macular degeneration, glaucoma, pentosan polysulfate sodium toxicity, and intraocular lenses. We have found the SNR to be sufficient; however, if necessary, it can be readily increased by increasing the integration time, with only a negligible impact on the AO bandwidth. For example, a 2× increase from 0.126 ms to 0.252 ms increases the total system latency of 3.1 ms by only 0.126 ms (or 4%).

In the control software, we substantially increased the data processing speed by parallelizing and optimizing the code. This enabled data processing to be completed in just 0.5 ms with an off-the-shelf workstation, at least 13× faster than previously reported. A particularly challenging issue with the control software was determining which AO control scheme would provide better performance: continuous or discontinuous exposure. We discuss this issue separately below.

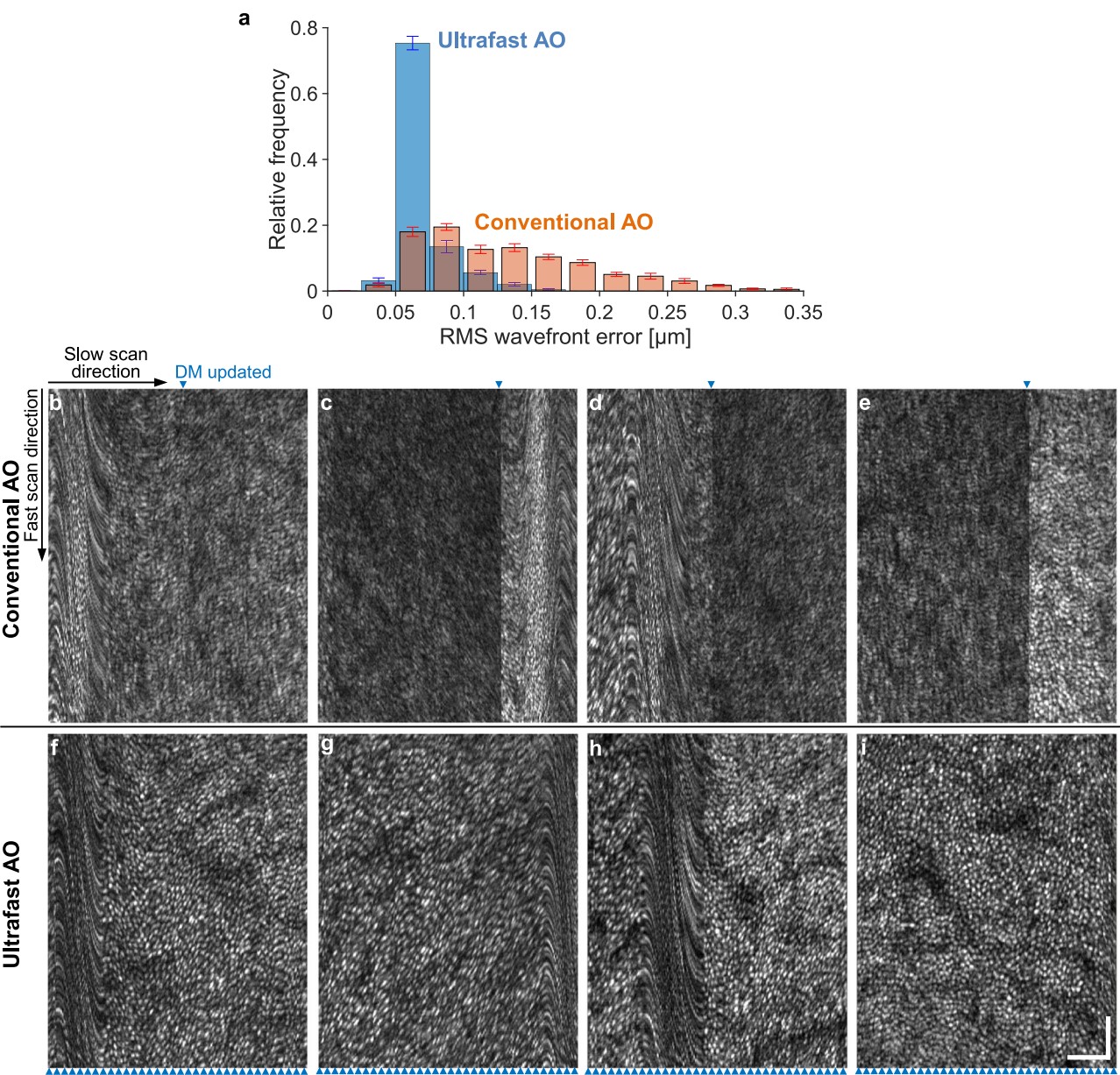

**Fig. 9 | Ultrafast ophthalmic AO provides more effective aberration correction and better clarity of cone mosaic in AO-OCT images acquired from a nystagmic eye.** The nystagmus caused the eye to jerk horizontally (along the fast scan axis, here displayed vertically). **a** Distribution of RMS wavefront error after aberration correction with conventional and ultrafast AO in the nystagmic eye. Error bars and their centers represent the standard error and mean of 15 videos. **b–e** Example frames in Supplementary Movie 5 where conventional AO failed to achieve adequate image quality. **f–i** Frames acquired by ultrafast AO with microsaccades occurred at similar locations as those with convention AO shown on top. No microsaccades in (**e**) and (**i**). Blue arrowheads denote the time points when the DM shape was updated. The AO-OCT volume acquisition rate was 6.7 Hz for a field of view of 1.3° H × 1° V. Because the AO loop rate for conventional AO was only slightly higher (10 Hz), the blue arrowhead position moves from frame to frame. Scale bars = 50 μm and apply to (**b–i**).

Although this study was conducted using 0.79 μm wavelength for wavefront sensing, the results are generalizable to other wavelengths because monochromatic aberrations of the eye other than defocus are insensitive to wavelength[72].

### Importance of AO rate-gain product

We have investigated two fundamental AO operational schemes: continuous and discontinuous exposure. The former is commonly used in astronomy and vision science and was previously used by our group. Some vision science groups have used the latter, but to the best of our knowledge, no literature describes its use in ophthalmic or astronomical AO. In this work, we experimented with both schemes because it was unclear which scheme would yield better performance for ophthalmic use. We found that the "slower" (in terms of AO loop rate and SHWS camera frame rate) discontinuous-exposure scheme achieves a larger AO bandwidth and a shorter AO convergence time (Fig. 1) than the "faster" continuous-exposure scheme. The larger AO bandwidth with discontinuous exposure results from a larger rate-gain product: we demonstrated both experimentally and theoretically that the AO bandwidth is a monotonically increasing function of the rate-gain product. It is a common misconception, at least in the ophthalmic field, that loop rate is the most important parameter to optimize for improving the temporal performance of an AO system. Perhaps because of this misconception, loop gain is reported much less

frequently in the literature. Based on our findings, we recommend optimizing the rate-gain product to improve AO temporal performance. Of course, if the AO system can have its gain set at or near 1, the task reduces to optimizing loop rate, as illustrated in our analysis in Fig. 4.

The optimal loop gain values obtained from a model eye (Fig. 1) represent upper bounds when applied to human eyes. In practice, we found that ultrafast AO with the discontinuous-exposure scheme was able to use the maximum loop gain of 1 in all demonstrated scenarios in this study and across the more than 20 subjects we have imaged to date with various retinal diseases and conditions (as mentioned above). By contrast, loop gains of 0.3−0.5 are commonly used with the continuous-exposure scheme[18,34−37,40,55,73−76]. This implies that the loop rate under continuous exposure would have to be 2−3.33 times higher than under discontinuous exposure to achieve a similar AO bandwidth.

The continuous-exposure scheme is restricted to a lower loop gain than the discontinuous-exposure scheme in order to maintain stability. This requirement arises because under discontinuous exposure the wavefront sensor always measures the DM correction calculated from the previous wavefront sensor image, whereas under continuous exposure with a two-frame delay[55] the current wavefront sensor image measures the DM correction calculated from the image two frames back. For continuous-exposure schemes with a smaller delay than two frames, the DM updates during the wavefront sensor exposure, resulting in the overlapping of two wavefront aberrations (before and after the DM updates) within a single frame. This overlap reduces the accuracy of the wavefront measurement[53].

## Respective benefits of the two fundamental AO operational schemes

In this work, we compared the two AO operational schemes using the same hardware. We found that the discontinuous-exposure scheme achieves a larger AO bandwidth. This scheme has other benefits compared with the continuous-exposure scheme. First, in AO systems that use separate beams for wavefront sensing and imaging[32,77], the wavefront sensing beam does not need to illuminate the eye during the data transfer, processing and DM actuation stages of the AO loop. This reduces light exposure to the eye and enhances subject safety. Second, discontinuous exposure allows more straightforward implementation than continuous exposure. The former requires only one software execution thread, while the latter generally requires two threads running concurrently—one for reading the images from the SHWS camera, the other for processing the SHWS images and controlling the wavefront correction device[53,54].

A key benefit of the continuous-exposure scheme over discontinuous exposure is its ability to capture more photons during the same time interval (less dead time), hence it is used by astronomical AO where the signal is usually much weaker than in ophthalmic AO. In astronomical AO, the continuous-exposure scheme also helps minimize aliasing of high-temporal-frequency aberrations by using an integration time similar to the AO loop period. However, this results in increased latency for AO systems that use a scientific CMOS (sCMOS) camera as a wavefront sensor, the type used in this study for its high sensitivity, speed, dynamic range, and field of view. The integration time must be increased so that it is at least as long as the readout time of the sensor (2.9 ms for the selected region of interest of our camera). This increases the total system latency by 2.8 ms in our case, which is significant compared to the original latency of 3.1 ms, resulting in reduced AO power rejection performance and AO bandwidth.

## Benefits of high-speed AO

Using the ultrafast ophthalmic AO system, we characterized ocular aberration dynamics in several targeted clinically-relevant scenarios such as absence of cycloplegia, eye blinks, displacement of a high-

power contact lens, change of fixation location, and nystagmus. In all cases, we observed aberration power spectra that were 1−2 orders of magnitude higher than those under normal laboratory conditions. Using the mean power spectra, we predict the minimum loop rates at which AO must operate in each scenario and find them to be 20−160 Hz (with a loop gain of 1), corresponding to bandwidths of 2.7−21.3 Hz. These bandwidths exceed the established 1−2 Hz bandwidth criterion that has guided the design of ophthalmic AO systems for the past two decades. Even higher bandwidths are likely needed in older or more severely diseased subjects, a population that is of significant clinical interest and may benefit the most from AO imaging. Characterizing the aberration dynamics in such population remains.

Compared with conventional AO (10 Hz), our ultrafast AO (233 Hz) provides an order of magnitude faster convergence after eye blinks and during sequential fixation, better correction through a contact lens in myopic and keratoconic eyes and in nystagmic eyes with fast eye movements, and more stable focus at a targeted retinal depth in non-cyclopleged eyes. Because ultrafast AO reduces the impact of eye blinks and eye motion, it enables near-continuous data acquisition for simpler and higher-throughput imaging in the clinic and in the laboratory.

Taken together, our results demonstrate the need for faster AO. We conclude that ultrafast ophthalmic AO is not only important for retinal imaging but can also be readily applied to the eye. With continued advancements in more capable and affordable cameras and computer technologies, we expect ultrafast ophthalmic AO to become a standard tool in laboratory and clinical settings where high-resolution retinal imaging, precise stimulation, and improved visual performance are required.

## Methods
### AO wavefront sensor and data processing algorithm
For our SHWS, we optimized the distance between the microlens array and the camera sensor by positioning the sensor at the geometric focus of the microlens array[78]. We enabled on-board 2 × 2 pixel binning of the Lightning camera in order to reduce the data transfer and processing workload, thereby reducing the latency. Even with 2 × 2 pixel-binning, the sampling of the sub-aperture image is comparable to that used in previous works[53,79], and we did not notice a degradation in AO performance nor retinal image quality compared with the no-binning case. Because the mimicked conventional AO used a much longer integration time (45 ms) than ultrafast AO (0.126 ms), we employed neutral density filters to ensure that the photon count on the SHWS accumulated during a 45 ms exposure matched that accumulated during 0.126 ms exposure for ultrafast AO. This ensured the same SNR for both conditions and avoided saturating the SHWS camera. Our experimental results presented in Supplementary Note 7 show that the effect of the ND filters on wavefront measurement is negligible.

We used a two-step thresholding center-of-gravity (TCoG) method to compute the centroid of each lenslet focal spot: First, we subtracted a calibration image (the fixed pattern noise, acquired with the lens cap on, plus a global threshold value) from the SHWS image. Second, the highest-intensity pixel for each sub-aperture was located[79]. Third, within a 11 × 11 super pixel square centered on each highest-intensity pixel, we subtracted an adaptive threshold consisting of 30% of the highest-intensity value. Finally, we computed the centroid of each SHWS spot as the center of gravity of the corresponding 11 × 11 super pixel square. The corrective voltages to be applied to the DM were determined using the direct slope control method and an integral controller scheme with no leak[1,35,73,80,81]. The control matrix contained 85 system modes, determined by performing singular value decomposition on the influence function matrix and removing the 12 modes with the smallest singular values. To monitor the AO performance in real-time, the wavefront was reconstructed from wavefront slopes

using 63 Zernike polynomial modes (up to the 10th radial order), and RMS wavefront error was computed from the Zernike coefficients. A blink is identified by 50% of the lenslet focal spots having peak intensities below a specified threshold (700). In this case, the DM holds the shape from the last non-blink frame.

### Integration into the Indiana AO-OCT system

We evaluated the performance of our ultrafast ophthalmic AO by integrating it into the Indiana AO-OCT system[58,59] and using this system to image the living human retina. The system uses a point-scanning spectral domain OCT subsystem that acquires A-scans at a rate of 1 MHz. The light source is a superluminescent diode (Superlum, Ireland) with a central wavelength of 790 nm and a bandwidth of 42 nm, which provides an axial resolution of 4.7 μm in retinal tissue ($n = 1.38$). The lateral resolution achieved with the ultrafast AO subsystem is 2.4 μm for a 6.7 mm eye pupil. Scanning was configured so that A-scans sample the retina at 1 μm/pixel in both lateral dimensions. The volume rate was 10 Hz for a 0.8° H × 1° V field of view. Power of the AO-OCT beam entering the eye was ~420 μW. Approximately 10% of the light reflected from the eye was directed to the SHWS with a Pellicle beamsplitter. We initially corrected the subject's sphero-cylindrical refractive error using the subject's prescription and the DM, before closing the AO loop to bring the subject within the dynamic range of the wavefront sensor. Once within the range, the AO control takes over. If the subject's prescription is unavailable, we empirically adjust the sphere and cylinder corrections with the DM until the RMS wavefront error is minimized. The subjects were directed to fixate on a target placed at their far point, achieved with a Badal optometer. All images were acquired at 3° temporal to the fovea, except for Fig. 5a, which was acquired at 2.4° temporal to the fovea to avoid big blood vessels. The procedures on the subjects adhered to the tenets of the Declaration of Helsinki and were approved by the Institutional Review Board of Indiana University. Written consent was obtained after the nature and possible risks of the study were explained. Consent was also obtained to publish the subject information shown in Supplementary Table 1.

### Measuring the temporal performance of the ultrafast ophthalmic AO system

The power rejection curve is the primary metric for characterizing the temporal performance of an AO system and is defined as the ratio of wavefront aberration PSDs with and without closed-loop AO correction[35,36,55,60]. We measured the power rejection curve by inserting a model eye into our AO-OCT system (with scanners on as when we do imaging) and applying a series of pseudo-random aberrations to the system's DM following Gofas-Salas et al.[55]; this approach ensures that the same input aberration is applied to the system both with and without closed-loop AO correction, which is critical for accurate measurement of the power rejection curve. The pseudo-random aberrations (P[n]) that we applied to the DM consisted of pink noise, whose PSD follows $1/f$[55].

While the DM displayed the same sequence of pseudo-random aberrations, we acquired two sequences of SHWS images at ~1000 time points. One sequence was with and the other was without closed-loop AO correction. We reconstructed the aberrated wavefront sequences from the two SHWS-image sequences. For each lenslet location in the aberrated wavefront, we calculated its PSD by dividing the squared magnitude of the discrete Fourier transform of the wave aberration sequence at that location by the spectral resolution (=1/total measurement time). We then averaged the PSDs of all used lenslets to obtain the average PSD with and without closed-loop AO correction. The ratio of these two PSDs gives the power rejection curve. For reference, our detailed method and code that generate the pseudo-random aberrations for each DM actuator are provided in Supplementary Note 8 and Supplementary Software.

### Statistics and reproducibility

The results shown in Figs. 2d−f, 5a, b, 6a, 7a, c, f, and 9b−i are representative. The experiments associated with each of these figures were repeated 5, 5, 3, 12, 12, 12, 12, and 15 times, respectively, with similar results.

### Reporting summary

Further information on research design is available in Nature Portfolio Reporting Summary linked to this article.

## Data availability

The data that support the findings of this study are presented in the paper and the Supplementary Information. The raw data is too large to be shared publicly but is available from the first author upon request. The expected timeframe for responding to access requests is within one month. Source data is provided with this paper and can be accessed in a public repository on Github (https://github.com/yanliulight/Source_data).

## Code availability

The code for calculating the theoretical power rejection curve and noise transfer function, as well as for generating the pseudo-random aberrations used for measuring the power rejection curve is provided as Supplementary Software, which can be accessed in public repository on Github (https://github.com/yanliulight/Supplementary_software). The code that supports the plots and images within this paper is available from the first author upon request.

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

## Acknowledgements

This work was supported by the National Eye Institute grants R01-EY018339 (D.T.M) and R01-EY029808 (D.T.M). The authors thank Drs. Stephen A. Burns, Donald Gavel, Alfredo Dubra, and Serge Meimon for helpful discussion, Tim Clarke for machining support, and Kristen Bowles-Johnson for referring a subject.

## Author contributions

Y.L. and D.T.M. designed the research. Y.L., D.T.M., K.K., and H.W.J. developed the hardware system. J.A.C. and Y.L. developed the control software. Y.L. performed the theoretical analysis. Y.L., K.K., M.T.B, Q.J., A.L., H.W.J, M.J.K. and D.T.M. performed the experiments. M.M. characterized the nystagmic subject. Y.L. processed the data and drafted the manuscript. All authors revised the manuscript. D.T.M. supervised the project.

## Competing interests

The authors declare no competing interests.
