## [Transparent Peer Review file · Nature Communications]

Ultrafast adaptive optics for imaging the living human eye

Corresponding Author: Professor Donald Miller

Version 0:

Reviewer comments:

Reviewer #1

(Remarks to the Author)

This is a very interesting study on OCT retinal imaging with high-speed AO. It questions the validity of current systems and argues for the need of higher speed. It shows this for a selected number of subjects and conditions. My specific comments are as follows:

Abstract: sufficiently fast ophthalmic AO is possible without sacrificing other performances. -> Ultimately this would be a question of signal-to-noise (photons that can be detected in a short time interval). Please comment. Also clarify, how do you know that it is "sufficiently" fast? Could some processes not be faster?

Introduction:

L. 32: to correct the unique optical defects in each individual eye? It does not correct defects, but aberrations. Defects such as drusen, opaque lens, etc, are not corrected. Please edit.

L. 58-59: why would keratoconus, or high myopia, require higher AO speed? These are static large aberrations.

Results:

L. 99: τ RMS wavefront error due to wavefront sensing noise is much lower than the diffraction limit -> specify a fraction, as "much lower" is not very well defined.

L. 101: Is there a specific reason why 342 Hz was chosen? With higher beam power of the sensing beam (above 40 μ W) could the speed have been increased further?

Fig. 1 shows what appears to be individual sets of data. Did you repeat it, and could you then add error bars to the data sets? If not, please comment on the variability with AO off and AO on.

L. 181: Is the optimal loop gain not dependent on the subject being imaged? Destabilization may happen with different loop gains for different subjects?

L. 186: A wavelength of 0.79 μ m was used, but if a different wavelength was used, could that improve further the AO speed?

L. 200: ($= 342 \times 0.45 = 153.9$)... use same number of digits as for the discontinuous-exposure case, so here the result should be 154 (not 153.9).

Fig. 2: Again with the 2 markers, it would be good to indicate error bars.

Can you comment on the upper limit of the wavefront order used? Is it to the 4th, 5th, 6th or 7th radial order?

Fig. 3: shows only a cone mosaic image for one of the subjects. For a fair comparison, it would be relevant to also show that of the two other subjects.

Fig. 4: Did you use a threshold in the spot detection (setting intensities = 0 below a certain threshold) or do you calculate

centroids with brightness from 0 to the maximum brightness?

Fig. 5 contains very impressive work on the direct comparison of different conditions, and why a high >100 Hz AO speed is needed. The subject numbers are in the hundreds, do you have more subjects with the same conditions, for example, nystagmus, to evaluate variations between subjects with the same conditions? Did you image more people wearing contact lenses, to know if the one you imaged is a representative example, or an outlier from the average with more subjects?

L. 423: focus depth is much more stable -> Be more precise, how much more stable? An order of magnitude?

L. 444: Ultrafast AO converges much faster after a blink -> again, be more precise with the use of the word "much".

L. 494: Ultrafast AO converges much faster after changing -> again, be more precise with the use of the word "much".

L. 610: provides much faster convergence -> again, be more precise with the use of the word "much".

It would be very useful to know more about other subjects, also with similar conditions, to know / understand the variability in the subjects and the results for the same conditions analyzed.

(Remarks on code availability)

Reviewer #2

(Remarks to the Author)

This paper summarized Univ. of Indiana efforts to develop and characterize a high-speed AO instrument. This paper is well-written and is of high interest to the research community. The work summarized here by the Indiana group is outstanding! A few aspects however need further consideration and potential improvements:

1. The ultrahigh speed is somewhat overstated: 30x improvement in AO bandwidth is indeed an important achievement; however, this is still not ultrahigh speed. I would recommend toning down a bit this claim.
2. It is stated that most of the AO systems have close loops that do not exceed 1.4Hz. However, work performed by Hammer et al. (Biomed Opt Express. 2022 Nov 1; 13(11): 5860–5878) reports an AO speed of >13Hz.
3. It is stated that "Several higher-speed ophthalmic AO systems have been developed and have reported improved correction performance and retinal image quality [36, 42-46]." The number of references here is rather small, as more systems have been reported to date, including clinical-friendly compact systems. The reference list needs to be updated.
4. The discontinuous exposure scheme is very interesting and potentially beneficial, as it seems to yield improved performance, especially higher AO loop gain. However, it is not very clear what happens if the patient blinks or loses fixation at a lower DM actuation speed. Has been this method tested on difficult patients, where the SNR tends to be lower than in normal subjects? The SNR of the WS may be problematic in these subjects.
5. In the supplementary material, video 1 (conventional AO) shows frames out of sync. That might need to be corrected, although it is irrelevant to this discussion.
6. The RMS wavefront error is much lower and the AO converges faster with the ultrafast AO. Would it also perform better than conventional AO in case of micro saccadic motion? I assume not, as this is not correctable.
7. Improvements in nystagmic eye are fabulous, where the eye exhibits large movements. Would a strip registration algorithm further improve the results in this case?

(Remarks on code availability)

Reviewer #3

(Remarks to the Author)

Summary

Liu et al. present a novel high-speed approach to measuring and correcting the optical aberrations of the eye using adaptive optics (AO) ophthalmoscopy. The study consists of a thorough theoretical evaluation, implementation of the high-speed AO into an AO-OCT system, and experimental validation on human subjects under a variety of potentially challenging imaging conditions. The data and analysis are quite convincing and support the authors' conclusions. The results show the potential to improve the quality of AO imaging and extend this technique to scenarios that may be challenging in certain clinical applications. While the clinical potential is demonstrated, the authors somewhat overstate the specific clinical significance of the imaging results obtained in this study. Furthermore, there are several technical details and points of discussion that could be clarified better and generalized to better connect the findings here to the rest of the field. Overall, these concerns are relatively minor relative to the significance of the results and high quality of the study and manuscript.

Major Concerns/Questions

1. Throughout the manuscript, the authors refer to the demonstrated application of the ultrafast AO to several "clinically-relevant" scenarios presented at the end of the results. While the results are impressive, and the demonstrated AO imaging of a patient with nystagmus does present an indication of clinical potential, the other examples do not strike me as particularly clinically relevant. For example, the need to image a patient wearing a contact lens or without cycloplegia is not necessarily something encountered routinely in clinical AO imaging. All of the examples demonstrate the benefits and improvement of the high-speed AO, but I would hesitate to categorize most of these as "clinically-relevant". It would be more accurate to describe the other demonstrations as illustrative examples or scenarios instead.

2. The loop gains used in the continuous-exposure scheme are significantly lower than in the discontinuous scheme. In the main text, the authors hint at the possibility that this is due to the fact that the deformable mirror (DM) update step is occurring simultaneously with wavefront sensor (WFS) exposure, affecting the measured aberrations. Is this the belief of the authors? It would be nice to have a more complete and explicit discussion of this in the main text.

3. The authors conclude that the discontinuous-exposure scheme is preferable due to the fact that its rate-gain product is larger than that of the continuous-exposure scheme. Is there any situation that could arise in which the continuous-exposure scheme would be beneficial relative to the discontinuous-exposure scheme? As the optimal loop gain was determined from a model eye, is it expected that the optimal loop gains will be identical in the human eye as well?

Minor Points

4. Line 43 – It may be a good idea to clearly define AO bandwidth in this first use and distinguish this parameter from AO loop rate for clarity.

5. Line 48 – What changes would you expect in clinical scenarios. A few examples here may be helpful.

6. Lines 49-55 – This section seems more well-suited for the discussion in its current form. Otherwise, if a common set of AO clinical requirements is to be assembled, it is likely best rooted with citations to demonstrate that this set of conditions represents the consensus of the field.

7. Line 77 – What are the anticipated highest frequency aberrations of the eye. This is hinted at, but not directly addressed.

8. Line 78 – “The entire temporal spectrum of ocular aberrations.” This seems a bit overstated and potentially an impossibility to measure.

9. Line 80 – Image quality is a bit vague. What is improved in the images?

10. Line 145 – It would be helpful to clarify the precise definition of latency used here.

11. Lines 179-190 – Were these experiments performed in a stationary model eye? Please clarify.

12. Line 218 – The expressions Herror and Hreject are used interchangeably in a few places and could lead to confusion. I'd recommend sticking to one for consistency.

13. Line 249 – “Figure 2d,f” should refer to figure 1. There are a few other instances of Figure 1 being referred to as Figure 2 as well. Check all in-text figure labels for consistency.

14. Pages 6-7 – There are several points where the reader jumps back and forth between Figure 1 and Figure 2. Given that Figure 2 is a single plot that appears somewhat connected to the message of Figure 1, it may make sense to combine these into a single figure.

15. Line 296 – I was looking for this table much earlier while reading. I would highly recommend moving this a few pages forward in the manuscript, possibly as early as page 3.

16. Figure 5a,c – As noted in major concern 1, describing and grouping these different imaging conditions as “clinically relevant” is not an accurate characterization of what was performed here. Removing this label from the plot and just listing the imaging conditions would be preferable.

17. Line 414-442 – In most cases, AO imaging (including in clinical settings) is performed after cycloplegia. The results the authors show in this section are impressive, but may be a complicated solution to a problem that could be more easily solved by simply using cycloplegic drops. The authors should better address the motivation for this approach.

18. Lines 445-454 – The discussion on tear film seems a bit oversimplified and does not consider the possibility of tear-film breakup. While changes in the tear-film may be correctable, the situation will be much more challenging in cases where the tear-film may become a discontinuous surface on the cornea. Furthermore, suggesting that this will improve performance in those with dry eye seems a bit speculative. Dry eye seems likely to present more problems for wavefront sensing than beyond increased blink frequency and higher frequency aberrations.

19. Figure 7 – For consistency, it would be nice to see the conventional vs. ultrafast AO images for panel c similar to the format of panels a and b

20. Lines 603-605 – The authors predict the minimum loop rates for the imaging scenarios described (assuming loop gain of 1). Given the emphasis placed on rate-gain product throughout the manuscript, it would be more logical to state these parameters in this more generalized form.

21. Lines 629-633 – Did the authors notice any effect of the filters on the measured wavefront? Do the filters have a noticeable effect on RMS wavefront error?

22. Lines 656-657 – The procedure for correcting for the subject's spherocylindrical error should be expanded here. Was this done based on the subject's prescription? Empirically based on the AO wavefront sensor? If done empirically, was this an objective measure (minimize RMS error)? Or subjective (sharpest spots)?

Supplemental

23. Figure S1 – The authors only briefly discuss the treatment of their camera with rolling shutter as a global shutter. More generally, are their significant differences between expected outcomes using a rolling vs. global shutter? Is one preferable? What are the tradeoffs?

24. Supplementary Note 1 – While some of the transfer functions have fairly obvious forms (zero-order hold and delay), others are not as obvious to me (e.g. HWFS). It would be helpful to describe the origins of these equations or cite appropriate studies using this model.

25. Similarly, the choice of integrator controller is different than what I've seen in the past for theoretical analyses of AO control in retinal imaging systems (typically modeled as (loop gain)/s). Where does the Hcontroller expression derived here come from? How does this choice of controller affect the results vs. other choices?

26. Lines 146-157 – What was the loop gain value used here?

27. Line 264 – Similar to comment 20 above, instead of specifying the minimal AO loop rate, it may be more general to specify this as the rate-gain product.

(Remarks on code availability)

Version 1:

Reviewer comments:

Reviewer #1

(Remarks to the Author)

The authors have done an excellent job in revising the manuscript and have carefully addressed all of the questions and comments raised. As a result the manuscript is stronger and will be even more beneficial for the optics community.

(Remarks on code availability)

Reviewer #2

(Remarks to the Author)

The authors have adequately addressed all concerns and suggestions. I propose that the manuscript shall be accepted with the implemented edits.

(Remarks on code availability)

Reviewer #3

(Remarks to the Author)

The authors have done excellent work in responding to my and the other reviewer's points. The revised manuscript includes more supporting data across more clinically significant imaging scenarios which further supports the key points of the manuscript. Overall, the revised manuscript is very strong and clearly addresses all of the concerns raised.

(Remarks on code availability)

Response to review comments

We would like to thank the editor and all three reviewers for their critical reading of our manuscript and their constructive feedback. We have added new experimental results, performed additional analyses, and revised both the manuscript and supplementary materials to improve the quality and presentation of the work.

In addition to these requested changes, we made minor changes to improve grammar, accuracy and readability, and to include missed citations. These changes together with those requested by the reviewers make the manuscript sounder without altering the interpretation of our results or our conclusions.

In the point-by-point response below, the reviewers' comments are indicated in **black**, our responses are indicated in **blue**, and our action and revisions in the manuscript are indicated in **red**.

Point-by-point response to reviewers' comments

Reviewer #1:

This is a very interesting study on OCT retinal imaging with high-speed AO. It questions the validity of current systems and argues for the need of higher speed. It shows this for a selected number of subjects and conditions. My specific comments are as follows:

Abstract: sufficiently fast ophthalmic AO is possible without sacrificing other performances. -> Ultimately this would be a question of signal-to-noise (photons that can be detected in a short time interval). Please comment. Also clarify, how do you know that it is "sufficiently" fast? Could some processes not be faster?

Response: Thank you for the two questions. For Question 1, our abstract text "without sacrificing other performances" is in reference to three performance points that we discuss in the main text, only one of which is signal-to-noise. These points are

(1) The same high spatial performance of our previous AO system (97-actuator ALPAO DM and SHWS with a 20 × 20 microlens array sampling a 6.7 mm eye pupil).

(2) The same wavefront sensing beam power injected into the eye and then subsequently deflected into the SHWS.

(3) The wavefront error due to noise is below the diffraction limit, that is, photon and camera noise have negligible effect on AO performance. Three sub-sections elaborated this point:

- "Noise level of our ultrafast ophthalmic AO system has negligible effect on AO performance"
- The 4th paragraph in the Section "Characterizing ocular aberration dynamics up to 171 Hz in clinically-relevant scenarios and demonstrating the need for faster AO in these scenarios"
- "Supplementary Note 4: Determining the noise performance of our ultrafast AO system in terms of its residual wavefront error and noise transfer function"

While we agree with the reviewer that these performance points could be made clearer in the abstract by expanding on them, the word limit of the abstract precludes us from doing so. However, we note that point 3 on signal-to-noise is described separately in the main text from points 1 and 2, and this may contribute to confusion.

To address this concern, we have described these three points together in the 2nd paragraph of Section "Our ultrafast ophthalmic AO system" on Page 3 of the revised manuscript:

"We maximized the SHWS speed using off-the-shelf components without sacrificing the high spatial performance of our previous AO system [58, 59], which featured a 97-actuator DM (DM97-15 high speed, ALPAO). We also maintained sufficient signal-to-noise ratio (SNR) of the SHWS to ensure that photon and camera noise did not affect AO performance (discussed later). The same beam power used by our previous system illuminated the eye (~420 μW at 790 nm) and the same fraction (~10%) deflected into the SHWS."

For Question 2 (How do you know that it is "sufficiently" fast? Could some processes not be faster?), the answer is more complex than our text suggests as it depends on how significant the high-frequency aberrations are, how well the aberrations follow a power law at high frequencies, and

ultimately whose eye we measure (healthy, diseased, etc.), as well as the intended use of the measurement. Given these issues, we feel it would be better to moderate our text on this topic.

We have changed “We demonstrate that sufficiently fast ophthalmic AO is possible without sacrificing other performances” to “We demonstrate that this much faster ophthalmic AO is possible without sacrificing other system performances.”

Moreover, in the 2nd to last paragraph of the Introduction section on Page 2, we have changed “Because the temporal performance of our system is designed to surpass the fastest anticipated temporal aberrations in the human eye, it allows us to characterize, for the first time, the entire temporal spectrum of ocular aberrations in diverse clinically-relevant scenarios.” to “Because the temporal performance of our system vastly surpasses that of any ophthalmic system that we know of, it allows us to better characterize and correct the temporal spectrum of ocular aberrations in a wider range of real-life scenarios that are encountered in clinical settings.”

Introduction:

L 32: to correct the unique optical defects in each individual eye? It does not correct defects, but aberrations. Defects such as drusen, opaque lens, etc, are not corrected. Please edit.

>we state “optical defects” not “defects.” We just need to make this more explicit.

Response: The term ‘optical defect’ is commonly used in the literature to refer to aberrations or wavefront aberrations of an optical system. This is our use here. To make clear that we are referring to the wavefront aberrations generated by the eye’s optics, we changed “to correct the unique optical defects in each individual eye” to “to correct the unique optical defects (wave aberrations) in the cornea and crystalline lens of each individual eye”.

L. 58-59: why would keratoconus, or high myopia, require higher AO speed? These are static large aberrations.

Response: We agree that keratoconus and high myopia can be considered static aberrations. However, even a fixating eye is continuously moving, causing the static aberration pattern to continuously move relative to the imaging beam and generate temporal aberrations. Additionally, our scenarios include contact lens correction. The movement of the contact lens relative to the cornea introduces additional temporal aberrations. Therefore, the combined effect of eye and contact lens motion, along with the larger aberrations in keratoconic and myopic eyes, may necessitate a higher AO speed for correction, which we investigate.

To better clarify, we have changed “... and can cause abnormal optics requiring contact lens correction (e.g. keratoconus and high myopia <-6 diopters (D))” to “... and can cause abnormal optics requiring correction by contact lens (e.g., keratoconus [50] and high myopia), which can move about on the eye especially after a blink.”

Results:

L. 99: The RMS wavefront error due to wavefront sensing noise is much lower than the diffraction limit -> specify a fraction, as "much lower" is not very well defined.

Response: We agree that the term 'much lower' is not well defined, but we prefer not to elaborate on it in this introductory section. Instead, we address this issue in detail in Sub-sections 4 and 5, which rigorously evaluate the RMS wavefront error due to wavefront sensing noise and provide specific answers.

However, to avoid any potential confusion, we have removed the word 'much' here. Also, please note that our RMS wavefront error due to noise increased slightly (from 16 nm to 23 nm, see Supplementary Figure 3) when we expanded the study to include many more subjects, including 5 keratoconus and 5 high-myopic subjects wearing contact lenses. We suspect that this increase results from the contact lenses reflecting and scattering more of our 790 nm light.

L. 101: Is there a specific reason why 342 Hz was chosen? With higher beam power of the sensing beam (above 40 uW) could the speed have been increased further?

Response: The 342 Hz is the maximum frame rate of our wavefront sensing camera for a 1920 x 1840 pixel region of interest. This limit is due to the readout speed of the camera sensor, not the signal-to-noise ratio of wavefront sensing, as mentioned in our text "The wavefront sensor camera had a readout speed of 1.445×10^9 pixels/s, 3–100× higher than that used in the field [32-34, 36, 37, 55]. As a result, the camera achieved a maximum frame rate of 342 Hz".

As such, even if we increase the power of the wavefront sensing beam, the AO speed will not increase further.

Fig. 1 shows what appears to be individual sets of data. Did you repeat it, and could you then add error bars to the data sets? If not, please comment on the variability with AO off and AO on.

Response: Thanks for pointing out this shortcoming. We did repeat the experiments and have added error bars to the data sets in the revised Fig. 1 (see below). Error bars denote the standard deviation of three measurements. Note that most error bars (especially those in e and f) are smaller than the data markers, showing the high repeatability of the measurements.

L. 181: Is the optimal loop gain not dependent on the subject being imaged? Destabilization may happen with different loop gains for different subjects?

Response: The reviewer is correct that the optimal loop gain in theory depends on the subject, but in practice we found that this dependence is minimal for our system. To clarify this issue, we have added the following sentences in the second paragraph of the Discussion section “Importance of AO rate-gain product”.

“The optimal loop gain values obtained from a model eye (Figure 1) represent upper bounds when applied to human eyes. In practice, we found that ultrafast AO with the discontinuous-exposure scheme was able to use the maximum loop gain of 1 in all demonstrated scenarios in this study and across the more than 20 subjects we have imaged to date with various retinal diseases and conditions (as mentioned above).”

L. 186: A wavelength of 0.79 μm was used, but if a different wavelength was used, could that improve further the AO speed?

Response: The monochromatic aberrations of the eye (other than defocus) are largely wavelength independent [77], thus selecting a different wavelength for the SHWS shouldn’t improve or diminish our ability to measure the aberrations of the eye or control the speed of our AO, assuming that we maintain sufficient signal-to-noise. In our case, the AO speed is limited by the readout speed of the SHWS camera and the data processing speed of our software. We selected the SHWS wavelength to be 0.79 μm as that was the wavelength used for the retinal imaging part of the AO-OCT system,

thus simplifying system design. To clarify this issue in the manuscript, we have added the following paragraph in the Discussion section on Page 20:

“Although this study was conducted using a single light wavelength (0.79 μm) for wavefront sensing, the results are generalizable to other wavelengths because monochromatic aberrations of the eye other than defocus are insensitive to wavelength [77]”.

Ref. [77]: Marcos S, Burns SA, Moreno-Barriusop E, Navarro R, "A new approach to the study of ocular chromatic aberrations," Vision Res. 39, 4309-4323 (1999).

L. 200: ($= 342 \times 0.45 = 153.9$)... use same number of digits as for the discontinuous-exposure case, so here the result should be 154 (not 153.9).

Response: Thank you for pointing this out! We agree that we should be consistent in using the same number of digits. We prefer to use one decimal place for the loop gain in both schemes and therefore updated the product for the discontinuous-exposure case to $233 \times 1.0 = 233.0$ and kept the product for the continuous-exposure case as $342 \times 0.45 = 153.9$.

Fig. 2: Again with the 2 markers, it would be good to indicate error bars.

Response: Thanks for your suggestion! We have added error bars in Fig. 2 of the revised manuscript (also shown below), and updated the figure caption accordingly.

The error bars represent the standard deviation of three measurements. Please note that the variability is so small that the error bars are smaller than the data markers.

Can you comment on the upper limit of the wavefront order used? Is it to the 4th, 5th, 6th or 7th radial order?

Response: To monitor the AO performance in real-time, the wavefront was reconstructed from wavefront slopes using 63 Zernike polynomial modes (up to the 10th radial order and excluding piston, tip and tilt), and RMS wavefront error was computed from the Zernike coefficients. This

information was described in the sub-section “AO wavefront sensor and data processing algorithm” of the Methods section of the original manuscript.

We chose 10th radial order because correction of Zernike orders up through at least the 10th order is necessary to reach diffraction-limited imaging in ~95% of the population for a 7.5 mm pupil (see D. T. Miller and A. Roorda, "Adaptive optics in retinal microscopy and vision," Handbook of optics 3, 15.11-15.30 (2009)).

Also for the AO control matrix, we have added “The control matrix contained 85 system modes, determined by performing singular value decomposition on the influence function matrix and removing the 12 modes with the smallest singular values” in the same sub-section mentioned above.

Fig. 3: shows only a cone mosaic image for one of the subjects. For a fair comparison, it would be relevant to also show that of the two other subjects.

Response: Sure. We originally did not include those images in order to save space and keep the figure to a single column rather than a double column. As requested, we have added the cone mosaic images for the other two subjects in Fig. 3 of the revised manuscript (also shown below).

Fig. 4: Did you use a threshold in the spot detection (setting intensities = 0 below a certain threshold) or do you calculate centroids with brightness from 0 to the maximum brightness?

Response: We use a two-step thresholding center-of-gravity (TCoG) method to compute the centroid of each lenslet focal spot. This information was described in the sub-section “AO wavefront sensor and data processing algorithm” of the Methods section of the original manuscript.

Fig. 5 contains very impressive work on the direct comparison of different conditions, and why a high

>100 Hz AO speed is needed. The subject numbers are in the hundreds, do you have more subjects with the same conditions, for example, nystagmus, to evaluate variations between subjects with the same conditions? Did you image more people wearing contact lenses, to know if the one you imaged is a representative example, or an outlier from the average with more subjects?

Response: The subject numbers listed in the manuscript are the subject IDs in our university Institutional Review Board (IRB) system. In our original submission, we imaged one subject for each of the five clinically-relevant scenarios, as the goal was to test whether faster AO was beneficial in these scenarios. **Results acquired on those five subjects (Figs. 5–9) collectively showed the need for faster AO.**

While these results are encouraging, we agree with the reviewer that our study would be sounder if more subjects with the same conditions were examined. Hence, we performed additional experiments. We conducted 35 measurements across 24 subjects, covering six clinically-relevant scenarios along with one control scenario. **Five subjects were measured for each scenario.** The sixth clinically-relevant scenario is new and examined keratoconus subjects wearing a contact lens. The power spectra of ocular aberrations for each subject (35 measurements across the seven scenarios) are shown below and presented as Supplementary Figure 2 in the revised manuscript. We also updated Figure 5a,c using the mean of the five subjects for each scenario.

Supplementary Figure 2. Power spectra of ocular aberrations for each subject in each scenario, co-plotted with diffraction-limit (diff-limit) threshold for conventional and ultrafast AO. Five subjects were measured for each scenario. Error bars denote standard errors (15 repeated measurements for each nystagmic eye, 10 repeated measurements for each eye in all other scenarios). Subject information is given in Supplementary Table S1. Note that Subject S15 has a small pupil size (4 mm), which resulted in a lower power spectrum than those of other four keratoconus subjects in Supplementary Figure S2c. Also note that 27 out of 30 power spectra in clinically-relevant scenarios exceed the diffraction-limit threshold for conventional AO, highlighting the need for higher AO speeds than that provided by conventional AO.

From these new measurements, we observe some variations between subjects for the same scenario. However, **27 out of 30 cases (= 5 subjects per scenario × 6 scenarios) measured in the six clinically-relevant scenarios require higher AO speeds than that provided by conventional AO**, because the power spectra for these 27 cases are all above the diffraction limit threshold of conventional AO (black dashed line in the figure).

As the reviewer suggested, we avoided selecting outliers as representative examples in the main text of the revised manuscript by using the data from the subject whose power spectrum is the median of the five subjects measured in each clinically-relevant scenario. Note that in the nystagmus and high-myopia scenarios – in our first submission, our original subject turned out to be the median subject, so the plots for these conditions are unchanged. However, in the other three scenarios (without cycloplegia, with artificial tears, and during sequential fixation), our original subjects had above-median aberration power spectra. Hence, we have replaced those results with the data from the median subjects in those scenarios in the updated Figs. 6 – 8 of the revised manuscript. The conclusion remained the same: ultrafast AO substantially improves aberration correction and retinal image quality in these clinically-relevant scenarios compared to conventional AO.

L. 423: focus depth is much more stable -> Be more precise, how much more stable? An order of magnitude?

Response: Thank you for the suggestion. By analyzing the focus fluctuation across volumes in Supplementary Movie 1, we found that the focus depth is an order of magnitude more stable with ultrafast AO. Hence, we have added “an order of magnitude” before “more stable” in that line.

L. 444: Ultrafast AO converges much faster after a blink -> again, be more precise with the use of the word "much".

Response: We quantified improvement in AO convergence for each of the four scenarios in this sub-section, finding ultrafast AO converged at least an order of magnitude faster than conventional AO in all scenarios (14x, 11x, 10x and 17x faster). Following the reviewer’s recommendation, we have changed “much faster” to “an order of magnitude faster” in the sub-section title.

L. 494: Ultrafast AO converges much faster after changing -> again, be more precise with the use of the word "much".

Response: Thank you. We have changed “much faster” to “an order of magnitude faster.”

L. 610: provides much faster convergence -> again, be more precise with the use of the word "much".

Response: Thank you. We have changed “much faster” to “an order of magnitude faster.”

It would be very useful to know more about other subjects, also with similar conditions, to know / understand the variability in the subjects and the results for the same conditions analyzed.

Response: Thank you. As detailed in our response to the reviewer’s previous comment, we have performed additional experiments to better understand inter-subject variability. For each of the original five clinically-relevant scenarios, we have measured five subjects instead of just one. Additionally, we introduced a sixth clinically-relevant scenario examining keratoconus, which also included five subjects. In total, we conducted 30 subject measurements for the clinically-relevant scenarios, 6× more than the five measurements in the first submission.

Reviewer #2:

This paper summarized Univ. of Indiana efforts to develop and characterize a high-speed AO instrument. This paper is well-written and is of high interest to the research community. The work summarized here by the Indiana group is outstanding!

Response: Thank you for your positive feedback on our work!

A few aspects however need further consideration and potential improvements:

1. The ultrahigh speed is somewhat overstated: 30x improvement in AO bandwidth is indeed an important achievement; however, this is still not ultrahigh speed. I would recommend toning down a bit this claim.

Response: We appreciate the reviewer's concern. The term 'ultrafast' is commonly used in scientific literature to describe an increasingly wide range of processes or phenomena that occur over very short durations or at very high speeds, often on a scale tailored to the application. Google Scholar found ~1 million papers that used 'ultrafast'. While we are not wedded to the term, we feel it (1) captures the 30x improvement in bandwidth of our AO system compared to other systems in the ophthalmic AO field and (2) facilitates performance comparison to current ophthalmic AO systems, which we refer to in the manuscript as 'conventional.' Also, 'ultrafast' is a simple term that readily conveys the importance of speed. Given these, we prefer to keep the term.

To avoid confusion about whether our use of 'ultrafast' is specific to ophthalmic AO systems, we reviewed each instance in the manuscript and added 'ophthalmic' where the context was unclear. We also included 'ophthalmic' in all section titles and figure captions that used the term 'ultrafast.'

2. It is stated that most of the AO systems have close loops that do not exceed 1.4Hz. However, work performed by Hammer et al. (Biomed Opt Express. 2022 Nov 1; 13(11): 5860–5878) reports an AO speed of >13Hz.

Response: We believe there is some confusion as to what the '>13 Hz' refers to in the Hammer et al. paper. The 13 Hz refers to their system's OCT volume rate, which is independent of their system's AO loop speed and AO bandwidth. Unfortunately, the authors did not report their system's AO bandwidth and did not describe enough system performance parameters (such as latencies, AO loop rate, AO loop gain) for us to reliably estimate it. From personal communication with the authors, we know that their AO loop gain is 0.3 and their AO loop rate is lower than their wavefront sensor camera speed, but we don't know how much lower. In the best scenario, their AO bandwidth is estimated to be smaller than 4 Hz, which is an order of magnitude lower than the AO bandwidth of our system (38.0 Hz).

Please note that we cited the paper (Ref. 55, E. Gofas-Salas et al., Appl. Opt. 57, 5635-5642 (2018)) that reported a measurement of 4 Hz AO bandwidth in Table 1. While we could not cite Hammer et al. (Biomed Opt Express. 2022) in Table 1 because the paper does not report a measurement of AO bandwidth and did not describe enough system performance parameters, we did add the citation of Hammer et al. (Biomed Opt Express. 2022) in response to your next comment.

3. It is stated that “Several higher-speed ophthalmic AO systems have been developed and have reported improved correction performance and retinal image quality [36, 42-46].” The number of references here is rather small, as more systems have been reported to date, including clinical-friendly compact systems. The reference list needs to be updated.

Response: We are sorry that the sentence did not properly state how we selected the cited papers. The papers had to report an AO loop rate at or exceeding 30 Hz and report correction performance or quality of the retinal image. We chose “at or exceeding 30 Hz” as it extends what we describe two paragraphs before where we state, “... typically not exceeding 30 Hz.”

We rechecked the literature, including those on compact systems, and found only one additional paper (Hammer et al. (Biomed Opt Express. 2022)) that matched this criterion. We have added the citation of this paper in the revised manuscript.

4. The discontinuous exposure scheme is very interesting and potentially beneficial, as it seems to yield improved performance, especially higher AO loop gain. However, it is not very clear what happens if the patient blinks or loses fixation at a lower DM actuation speed. Has been this method tested on difficult patients, where the SNR tends to be lower than in normal subjects? The SNR of the WS may be problematic in these subjects.

Response: Please note that in this study, both the discontinuous-exposure and the continuous-exposure schemes were set to **have the same exposure time**, resulting in the same SNR (signal-to-noise ratio) for wavefront sensing. As such, if the SNR for wavefront sensing is problematic for a particular subject, it will affect both schemes.

In our study, we imaged 25 subjects across six clinically-relevant scenarios and found the SNR to be sufficient in all cases. In three of these scenarios, the subject purposely blinked during the AO-OCT video acquisition. We observed that ultrafast AO performed better after a blink compared to conventional AO (see Fig. 7). During a blink (identified when 50% of the lenslet focal spots have peak intensities below a certain threshold), the DM holds the shape from the last non-blink frame, as described in the Methods section.

Since submitting the original manuscript, we have used the ultrafast AO system to image over 20 subjects with various ocular conditions and retinal diseases. We have found the SNR to be sufficient in all cases.

We have added this discussion on SNR as a paragraph to the sub-section “Benefits of high-speed AO” in the Discussion section on Page 20:

“We imaged 25 subjects across six clinically-relevant scenarios in this study and found the SNR to be sufficient in all cases, that is, the wavefront error due to noise was $< \lambda/14$. We have also used the ultrafast AO system to image over 20 subjects with various retinal diseases and eye conditions, including retinitis pigmentosa, age-related macular degeneration, glaucoma, pentosan polysulfate sodium toxicity, and intraocular lenses. We have found the SNR to be sufficient; however, if necessary, it can be readily increased by increasing the integration time,

with only a negligible impact on the AO bandwidth. For example, a 2× increase from 0.126 ms to 0.252 ms increases the total system latency of 2.846 ms by only 0.126 ms (or 4%).”

5. In the supplementary material, video 1 (conventional AO) shows frames out of sync. That might need to be corrected, although it is irrelevant to this discussion.

Response: This appearance in the frames is actually attributed to the AO-OCT video and DM activation being out-of-sync with each other. In conventional AO, the DM activation rate (AO loop rate) and the AO-OCT volume rate are both set at 10 Hz, and in this case has a non-zero phase delay between the two. Specifically, AO updates occur within the OCT volume rather than at the beginning or end of each volume. We intentionally did this to help readers more easily observe the jump in system focus due to insufficient AO bandwidth, as shown in Fig. 6a and Supplementary Movie 1.

To make it clearer, we added the following explanation in the caption of Fig. 6: “Note that AO updates within the OCT volume instead of at the beginning or end, thus the jump of system focus for conventional AO (indication of insufficient AO bandwidth) is more clearly seen in each image.”

6. The RMS wavefront error is much lower and the AO converges faster with the ultrafast AO. Would it also perform better than conventional AO in case of micro saccadic motion? I assume not, as this is not correctable.

Response: This is a challenging question. A micro saccadic motion generates two main errors, one a translation of the retina and the other a translation of the eye’s optics. The first (retina translation) requires only a tip/tilt correction by AO, but this is not correctable with AO as the AO doesn’t measure retina motion. Fortunately, retina motion can be correctable in post processing using image registration. In contrast, aberrations induced by translation of the eye’s optics are correctable by AO, if the AO speed is high enough.

This gets to the reviewer’s question: How fast does the AO speed need to be? Here is a back-of-the-envelope calculation. According to Martinez-Conde, et al (Table 3, 2004) [1], the mean speed of a micro saccade in human is roughly 10 to 120 degree/second at the retina. Assuming a typical isoplanatic patch size of the human eye (1 degree), this means the AO needs to have a bandwidth of at least 10 to 120 Hz to correct aberrations created by the mean micro saccadic motion. The bandwidth of our ultrafast ophthalmic AO system is 38 Hz, and therefore, our system should be able to correct an appreciable fraction of the aberrations as they are generated by the micro saccade but clearly not all. Despite the need for even faster AO, a major benefit of our ultrafast AO over conventional AO is the high image quality that it achieves immediately after the micro saccade occurs, which is clearly evident in the cone mosaic examples in Fig. 9.

This is a very interesting question, but we feel goes beyond the key points we want to emphasize with the nystagmic eyes. Given that the manuscript is already long, we prefer not to include this discussion.

Reference:

[1] S. Martinez-Conde, et al., "The role of fixational eye movements in visual perception," *Nat. Rev. Neurosci.* **5**, 229-240 (2004).

7. Improvements in nystagmic eye are fabulous, where the eye exhibits large movements. Would a strip registration algorithm further improve the results in this case?

Response: Thank you for the positive feedback on our result with the nystagmic eye. This is an interesting point. We believe that our high-speed AO system, by allowing us to see more cells in the retinal images, will enhance the effectiveness of our slice-wise image registration algorithm. This should increase the number of images that can be registered and improve the image quality of the averaged, registered images. However, the extent to which high-speed AO will improve registration is not yet known and will require further investigation.

Note that we did not pursue image registration in this study because our goal was to assess the impact of increased AO speed on image quality, which we were able to determine using individual raw images. Cone photoreceptor cells, for instance, were readily countable in single images without the need for registration (see Fig. 9).

Reviewer #3:

Summary

Liu et al. present a novel high-speed approach to measuring and correcting the optical aberrations of the eye using adaptive optics (AO) ophthalmoscopy. The study consists of a thorough theoretical evaluation, implementation of the high-speed AO into an AO-OCT system, and experimental validation on human subjects under a variety of potentially challenging imaging conditions. The data and analysis are quite convincing and support the authors' conclusions. The results show the potential to improve the quality of AO imaging and extend this technique to scenarios that may be challenging in certain clinical applications. While the clinical potential is demonstrated, the authors somewhat overstate the specific clinical significance of the imaging results obtained in this study. Furthermore, there are several technical details and points of discussion that could be clarified better and generalized to better connect the findings here to the rest of the field. Overall, these concerns are relatively minor relative to the significance of the results and high quality of the study and manuscript.

Response: We thank the reviewer for the positive feedback on our work and for the suggestions to improve the presentation of the work.

Major Concerns/Questions

1. Throughout the manuscript, the authors refer to the demonstrated application of the ultrafast AO to several “clinically-relevant” scenarios presented at the end of the results. While the results are impressive, and the demonstrated AO imaging of a patient with nystagmus does present an indication of clinical potential, the other examples do not strike me as particularly clinically relevant. For example, the need to image a patient wearing a contact lens or without cycloplegia is not necessarily something encountered routinely in clinical AO imaging. All of the examples demonstrate the benefits and improvement of the high-speed AO, but I would hesitate to categorize most of these as “clinically-relevant”. It would be more accurate to describe the other demonstrations as illustrative examples or scenarios instead.

Response: Thank you for your feedback. We are sorry that we did not sufficiently explain the clinical relevancy of our imaging scenarios. Clinical imaging sees a more diverse range of eyes under more challenging conditions and more stringent time constraints than are typically found in research studies. We refer to our targeted scenarios as “clinically relevant” because they are all applicable to clinic use, either because they include a disease or eye condition or because the imaging protocol improves ease of use, efficiency of imaging, or robustness of operation.

To make the connection more explicit, we have heavily edited the Introduction and then described in more detail the clinical relevancy of the six scenarios we investigated at the beginning of the section, “Characterizing ocular aberration dynamics up to 171 Hz in clinically-relevant scenarios and demonstrating the need for faster AO in these scenarios.” The new text in this section on Page 10 now reads:

“Because the temporal spectrum of ocular aberrations in clinically-relevant scenarios is poorly understood, we used the wavefront sensor of our ultrafast ophthalmic AO system to measure ocular aberration dynamics at high temporal resolution. We conducted 35 subject

measurements across 24 different subjects, covering six targeted scenarios, which we refer to as “clinically relevant,” along with one control scenario.

The six clinically-relevant scenarios we investigated were:

1. Normal eye with artificial tears. Eye drops are commonly administered to patients suffering from dry eye, a condition caused by tear deficiency or excessive tear evaporation and often secondary to other medical issues. These drops help maintain moisture on the corneal surface and are often used during retinal imaging examinations. However, the artificial tears that provide the greatest comfort are generally the most viscous, resulting in blurry vision and blurry retinal images, especially after a blink.
2. Normal eye without cycloplegia. A significant number of patients cannot use cycloplegic drops. This includes those with closed-angle glaucoma, an anterior chamber intraocular lens, or an occludable angle, as these conditions can impair aqueous humor outflow and pose serious risks. Also, cycloplegic drops are avoided in pregnant or nursing women due to potential health concerns for the fetus or infant. Many patients also find these drops uncomfortable and disruptive to their daily activities, as their effect can last for several hours.
3. Normal eye with sequential fixation. AO imaging is limited to the eye’s isoplanatic patch size ($\sim 1^\circ$) [71], which is much smaller than images acquired with clinical ophthalmoscopes ($>20^\circ$). Montaging — the integration of multiple images to expand the instrument’s effective field of view — is a common method to address the eye’s small isoplanatic patch size. However, this technique requires the subject (or instrument) to rapidly change fixation, which can introduce additional aberrations. In this scenario, we mimic the image acquisition process used in montaging.
4. Keratoconic eye with contact lens. Keratoconus is a progressive eye disease in which the cornea thins and takes on a conical shape, causing vision loss. It is a leading cause of corneal transplantation [50]. To improve vision, a hard contact lens is commonly used to help neutralize the aberrations of the irregularly-shaped cornea. However, the contact lens moves about on the eye, especially after a blink, causing extra temporal aberrations in addition to those caused by eye movement.
5. Myopic eye with contact lens. High myopia (≥ 6 diopters in magnitude) affects a significant portion of the global population (163 million people as of 2020) and is projected to affect 1 billion people (or 10% of the global population) by 2050 [72]. Unfortunately, standard methods for correcting refractive errors, such as using trial lenses or a Badal system in front of the eye, cannot be easily integrated into mirror-based AO imaging systems [44, 73]. Existing DMs still lack the dynamic range needed to correct high myopia. Contact lens correction thus offers a potentially attractive and clinically viable alternative for imaging these subjects by helping to conserve the dynamic range of the DM for correction of higher-order aberrations.
6. Nystagmic eye. Nystagmus is a condition characterized by involuntarily, rhythmic eye movements and may be associated with serious health issues, especially those affecting the brain. Due to the constant motion, imaging nystagmic eyes with AO retinal imaging systems is notoriously challenging [44, 74].”

Note that we added a new scenario on keratoconic eyes (#4 above), which was not in the first submission.

2. The loop gains used in the continuous-exposure scheme are significantly lower than in the discontinuous scheme. In the main text, the authors hint at the possibility that this is due to the fact that the deformable mirror (DM) update step is occurring simultaneously with wavefront sensor (WFS) exposure, affecting the measured aberrations. Is this the belief of the authors? It would be nice to have a more complete and explicit discussion of this in the main text.

Response: Thank you for this suggestion. We have added the following discussion in the Discussion Section on Page 21:

“The continuous-exposure scheme is restricted to a lower loop gain than the discontinuous-exposure scheme in order to maintain stability. This requirement arises because under discontinuous exposure the wavefront sensor always measures the DM correction calculated from the previous wavefront sensor image, whereas under continuous exposure with a two-frame delay [55] the current wavefront sensor image measures the DM correction calculated from the image two frames back. For continuous-exposure schemes with a smaller delay than two frames, the DM updates during the wavefront sensor exposure, resulting in the overlapping of two wavefront aberrations (before and after the DM updates) within a single frame. This overlap reduces the accuracy of the wavefront measurement [53].”

3. The authors conclude that the discontinuous-exposure scheme is preferable due to the fact that its rate-gain product is larger than that of the continuous-exposure scheme. Is there any situation that could arise in which the continuous-exposure scheme would be beneficial relative to the discontinuous-exposure scheme? As the optimal loop gain was determined from a model eye, is it expected that the optimal loop gains will be identical in the human eye as well?

Response: Thank you for these two questions. For the first question on the advantage of the continuous-exposure scheme, we have added the following paragraph in the Discussion Section on Page 21:

“A key benefit of the continuous-exposure scheme over discontinuous exposure is its ability to capture more photons during the same time interval (less dead time), hence it is used by astronomical AO where the signal is usually much weaker than in ophthalmic AO. In astronomical AO, the continuous-exposure scheme also helps minimize aliasing of high-temporal-frequency aberrations by using an integration time similar to the AO loop period. However, this results in increased latency for AO systems that use a scientific CMOS (sCMOS) camera as wavefront sensor, the type used in this study for its high sensitivity, speed, dynamic range and field of view. The integration time must be increased so that it is at least as long as the readout time of the sensor (2.9 ms for the selected region of interest of our camera). This increases the total system latency by 2.8 ms in our case, which is significant compared to the original latency of 3.1 ms, resulting in reduced AO power rejection performance and AO bandwidth.”

For the second question on the optimal loop gains, we have added the following discussion on Page 20 of the Discussion section:

“The optimal loop gain values obtained from a model eye (Figure 1) represent upper bounds when applied to human eyes. In practice, we found that ultrafast AO with the discontinuous-exposure

scheme was able to use the maximum loop gain of 1 in all demonstrated scenarios in this study and across the more than 20 subjects we have imaged to date with various retinal diseases and conditions (as mentioned above).”

Minor Points

4. Line 43 – It may be a good idea to clearly define AO bandwidth in this first use and distinguish this parameter from AO loop rate for clarity.

Response: Thank you for this suggestion. We have added definitions for both AO loop rate and AO bandwidth in the first sentence that uses them on Page 1. The sentence now reads:

“Most systems use wavefront sensor integration (or exposure) times of 10–60 ms with AO loop rates (frequency at which an AO system updates its correction) typically not exceeding 30 Hz, resulting in AO bandwidths (range of temporal frequencies over which an AO system effectively corrects aberrations) of ~1.4 Hz [32-43]”.

5. Line 48 – What changes would you expect in clinical scenarios. A few examples here may be helpful.

Response: The next paragraph presents AO speed requirements for clinical scenarios and includes examples, so we prefer to keep this paragraph intact. To aid the transition, the phrase, “(as described below)” has been added to this sentence, which has also been edited further to address other reviewer comments. The sentence now reads:

“Many of these scenarios could induce additional high- and low- temporal-frequency aberrations (as described below), yet almost nothing is known about the temporal properties of ocular aberrations in these situations — a striking gap given that at least half of current AO publications involve clinical research [3].”

6. Lines 49-55 – This section seems more well-suited for the discussion in its current form. Otherwise, if a common set of AO clinical requirements is to be assembled, it is likely best rooted with citations to demonstrate that this set of conditions represents the consensus of the field.

Response: This is a good point. Unfortunately, there is no established set of requirements for the clinical use of AO systems (at least for the purpose in which AO is used in this paper). However, there is a general understanding in the field as to what these requirements likely are. While this topic fits in the discussion (as suggested by the reviewer), we feel putting it up front in the paper is necessary for establishing the problem, without which we may lose the reader. It also reflects that ophthalmic AO remains an emerging field.

To address the reviewer’s concern, we have edited this section to (1) make it explicit that there are no established requirements for the clinical use of AO systems, but there is a general understanding of what these requirements likely are and (2) include citations.

The section now reads:

“There are no established criteria for clinical AO operation [4]. Nevertheless, there is a general understanding in the field that the AO system needs to be fast enough to permit clinicians and technicians to acquire high-resolution images easily, efficiently, and robustly under all kinds of eye conditions, similar requirements as those for operation of non-AO commercial ophthalmoscopes [44]. Specifically, a clinical AO system should: (1) quickly align to the patient; (2) allow rapid focus through and about the retina (traversing distances larger than the ocular isoplanatic patch and system’s depth of focus); (3) stabilize focus at a retinal depth of interest; (4) be robust against tear film disruptions, eye blinks, and eye and ophthalmic appliance (e.g., contact lens) motion; and (5) correct accommodation in non-cyclopleged eyes or in cyclopleged eyes but before or after peak cycloplegic effectiveness. Ensuring these five capabilities is more difficult in the presence of disease and aging, which could exacerbate conditions necessitating higher AO speeds. Disease and aging increase the incidence of dry eye [45] and nystagmus [46], reduce fixation stability [47, 48], prohibit cycloplegia in some patients (e.g. those with narrow-angle glaucoma) [49], and can cause abnormal optics requiring correction by contact lens (e.g., keratoconus [50] and high myopia), which can move about on the eye especially after a blink. Older eyes also exhibit increased aberrations [51], compounding the temporal effect of these conditions. These conditions likely necessitate higher AO speeds, but to what extent is unknown.”

7. Line 77 – What are the anticipated highest frequency aberrations of the eye. This is hinted at, but not directly addressed.

Response: The highest frequency could be infinite if the power spectral density continues to follow a power law at high frequencies. However, the power spectral density decreases exponentially with increasing frequency and above a certain point it becomes inconsequential. For example, assuming the power spectral density follows a power law, we can measure >99% of the total aberration power between [0.2 Hz, +∞) with our sampling rate of 342 Hz. Given that this analysis assumes a power law, which might not hold for all eye conditions, we feel it is better to moderate our text on this topic.

We have changed “Because the temporal performance of our system is designed to surpass the fastest anticipated temporal aberrations in the human eye, it allows us to characterize, for the first time, the entire temporal spectrum of ocular aberrations in diverse clinically-relevant scenarios.” to “Because the temporal performance of our system vastly surpasses that of any ophthalmic system that we know of, it allows us to better characterize and correct the temporal spectrum of ocular aberrations in a wider range of real-life scenarios that are encountered in clinical settings.”

8. Line 78 – “The entire temporal spectrum of ocular aberrations.” This seems a bit overstated and potentially an impossibility to measure.

Response: We agree, and toned down our assertion. The sentence now reads: “Because the temporal performance of our system vastly surpasses that of any ophthalmic system that we know of, it allows us to better characterize and correct the temporal spectrum of ocular aberrations in a wider range of real-life scenarios that are encountered in clinical settings.”

9. Line 80 – Image quality is a bit vague. What is improved in the images?

Response: We agree and have added clarification. The sentence now reads “We find that ultrafast AO significantly improves image quality (sharpness of retinal cells) and reduces wavefront error compared to conventional AO.”

10. Line 145 – It would be helpful to clarify the precise definition of latency used here.

Response: Thank you. We have added the definition of latency at its first use under Table 1: “Latency is defined as the time delay between the average exposure start time across rows of the SHWS camera and completion of the DM update [63, 64], indicated as T_{latency} in the timing diagram in Supplementary Figure 1.”

11. Lines 179-190 – Were these experiments performed in a stationary model eye? Please clarify.

Response: Yes, these experiments were performed in a stationary model eye. To clarify, the first sentence of this paragraph now includes:

“We measured the power rejection curve and bandwidth of our system under each of the two fundamental AO operational schemes, using a model eye and applying pseudo-random aberrations to the DM in the system (see Methods).”

12. Line 218 – The expressions Herror and Hreject are used interchangeably in a few places and could lead to confusion. I’d recommend sticking to one for consistency.

Response: Thank you! We have replaced the two instances of “Herror” with “Hreject” for consistency.

13. Line 249 – “Figure 2d,f” should refer to figure 1. There are a few other instances of Figure 1 being referred to as Figure 2 as well. Check all in-text figure labels for consistency.

Response: Thank you! We have made these corrections and re-checked all in-text figure labels.

14. Pages 6-7 – There are several points where the reader jumps back and forth between Figure 1 and Figure 2. Given that Figure 2 is a single plot that appears somewhat connected to the message of Figure 1, it may make sense to combine these into a single figure.

Response: Thanks for the suggestion. However, we prefer to keep the two figures separate for the following three reasons: 1) Figure 1 is already complicated and busy with 6 sub-panels; 2) Figure 2 involves a new concept of “rate-gain product”, which is introduced one page later in the main text after Fig. 1; and 3) Figure 1 spans two columns, while Fig. 2 spans only one, and thus appending Fig. 2 to Fig. 1 would create a blank space in the figure, which is not favorable.

15. Line 296 – I was looking for this table much earlier while reading. I would highly recommend moving this a few pages forward in the manuscript, possibly as early as page 3.

Response: Thank you for this recommendation. We agree and have moved Table 1 forward to the subsection titled “Our ultrafast ophthalmic AO system” on Page 3. The last sentence in this section now reads:

“See Methods for more details and Table 1 (column 2) for a summary of our measurements of key speed-related performance parameters of our ultrafast ophthalmic AO system.”

The two sentences at Lines 288-290 of the original manuscript have been reduced to one, which now reads:

“Table 1 (columns 2 and 3) compares key speed-related performance parameters of our ultrafast ophthalmic AO system with corresponding values for conventional ophthalmic AO.”

16. Figure 5a,c– As noted in major concern 1, describing and grouping these different imaging conditions as “clinically relevant” is not an accurate characterization of what was performed here. Removing this label from the plot and just listing the imaging conditions would be preferable.

Response: Following your suggestion, we have removed the label from the plot and just listed the imaging conditions in Fig. 5a, c. We also elaborated the clinical relevancy of our scenarios in the list on Page 10 in response to the reviewer’s first question.

17. Line 414-442 – In most cases, AO imaging (including in clinical settings) is performed after cycloplegia. The results the authors show in this section are impressive, but may be a complicated solution to a problem that could be more easily solved by simply using cycloplegic drops. The authors should better address the motivation for this approach.

Response: Thank you for pointing this out. Unfortunately a significant number of patients cannot use cycloplegic drops. This includes patients with closed-angle glaucoma, an anterior chamber intraocular lens, or an occludable angle, as these conditions can impair aqueous humor outflow and pose serious risks. Also, cycloplegic drops are typically avoided in pregnant or nursing women due to concerns that the drugs might affect the fetus or infant. Many patients also find these drops uncomfortable and disruptive to their daily activities, as their effect can last for several hours. Taken together, ultrafast AO extends the capability of AO for a significant group of patients. Also, while we agree that the engineering is complex, it offers what we believe is a straightforward solution for the clinician. Following the reviewer’s recommendation, we have added the above information as a paragraph in the list (#2) on Page 10 of the revised manuscript.

18. Lines 445-454 – The discussion on tear film seems a bit oversimplified and does not consider the possibility of tear-film breakup. While changes in the tear-film may be correctable, the situation will be much more challenging in cases where the tear-film may become a discontinuous surface on the cornea. Furthermore, suggesting that this will improve performance in those with dry eye seems a

bit speculative. Dry eye seems likely to present more problems for wavefront sensing than beyond increased blink frequency and higher frequency aberrations.

Response: We did not intend to oversimplify the dynamics of the tear film, but agree that clarification is needed. To address, we have added a new sentence and edited another one. The two sentences now read:

“This could be especially useful in older subjects whose tear films are unstable or in those suffering from dry eye, which requires frequent blinking. In such cases, ultrafast AO can improve image quality immediately after each blink, continuing until the tear film breaks, with the benefits diminishing thereafter until the next blink.”

19. Figure 7 – For consistency, it would be nice to see the conventional vs. ultrafast AO images for panel c similar to the format of panels a and b.

Response: We agree and have added the requested images in the revised Fig. 7g (also shown below).

After blink with high-viscosity artificial tears

20. Lines 603-605 – The authors predict the minimum loop rates for the imaging scenarios described (assuming loop gain of 1). Given the emphasis placed on rate-gain product throughout the manuscript, it would be more logical to state these parameters in this more generalized form.

Response: We agree with the reviewer’s sentiment, but the result here is not solely dependent on the rate-gain product. As indicated in Equation (1), the power rejection curve of an AO system depends on four parameters: $T_{\text{integration}}$, T_{DM} , T_{delay} , and rate-gain product. We were able to use the last, the rate-gain product, to explain the difference in performance between the continuous- and discontinuous-exposure schemes (Figs. 1 and 2 results) because we used the same AO system (wavefront sensor, DM, control computer, and data processing software) with both schemes, ensuring that the three timing parameters ($T_{\text{integration}}$, T_{DM} , T_{delay}) remained constant.

Based on the Fig. 1 and 2 results showing better performance with the discontinuous-exposure scheme, we used this scheme for the remaining studies in the manuscript, including those presented in Fig. 5. In Fig. 5c, we predict the temporal performance required for an AO system using the discontinuous-exposure scheme to achieve diffraction-limited performance for the subject

scenarios shown in Fig. 5a. The temporal performance of the AO systems is characterized by their simulated power rejection curves shown in Fig. 5b.

As explained in the first paragraph above, the power rejection curve of an AO system is influenced not only by the rate-gain product but also by the other three timing parameters ($T_{\text{integration}}$, T_{DM} , T_{delay}). The DM response time, T_{DM} , is hardware specific and thus not a readily adjustable parameter. However, $T_{\text{integration}}$ (wavefront sensor integration time) and T_{delay} (time delay due to pixel readout, data transfer and data processing) are adjustable, and are related to AO loop rate in our simulation for Fig. 5b, as we set $T_{\text{integration}} = T_{\text{delay}} = (1/\text{AO loop rate})/2$, following standard practice (i.e. a two-frame delay AO system [53, 55, 65]). $T_{\text{integration}}$ and T_{delay} affect the latency of the AO system and are independent of AO loop gain. Since we use the discontinuous exposure scheme in the simulated AO system, we set the loop gain to 1, consistent with our own system and other conventional AO systems that use a loop gain of 0.9 to 1 [32, 38]. Given that three ($T_{\text{integration}}$, T_{delay} , and rate-gain product) out of the four parameters that determine the power rejection curve are functions of the AO loop rate with the fourth parameter (T_{DM}) being a constant, AO loop rate is left as the primary adjustable parameter for simulating the power rejection curves of different AO systems using the discontinuous-exposure scheme, which we therefore use in Fig. 5b,c.

Considering these factors, we prefer to use the AO loop rate in this particular context.

21. Lines 629-633 – Did the authors notice any effect of the filters on the measured wavefront? Do the filters have a noticeable effect on RMS wavefront error?

Response: Thank you for raising this important question. We have added a new Supplementary Note where we present the experimental results that show the effect of the filters on wavefront measurement is negligible.

“Supplementary Note 7: Effect of the neutral density filters on the wavefront measurement is negligible.

Because the mimicked conventional AO used a much longer integration time (45 ms) than ultrafast AO (0.126 ms), we employed neutral density (ND) filters before the wavefront sensor to ensure that the photon count on the SHWS accumulated during a 45 ms exposure matched that accumulated during 0.126 ms exposure for ultrafast AO. This ensured the same signal-to-noise ratio for both conditions and avoided saturating the SHWS camera. Here, we present experimental results and show the effect of the ND filters on the wavefront measurement is negligible.

We mounted a model eye (composed of an achromatic lens and a business card as the retina) in the system and measured its aberration with and without using the ND filters. The exposure time was 45 ms with the ND filters and 0.126 ms without the ND filters, respectively. The wavefront measured with and without the ND filters are shown in Supplementary Figure 8a and 8b, and their difference is shown in Supplementary Figure 8c and 8d (with different scales).

Supplementary Figure 8. Effect of ND filters on the wavefront measurement. **a** Wavefront measured with the ND filters. **b** Wavefront measured without the ND filters. **c, d** Difference of the wavefronts measured with and without the ND filters, plotted in two different scales.

The difference of the two wavefronts is very small, with an RMS of $0.018 \mu\text{m}$, much smaller than the diffraction limit of $0.056 \mu\text{m}$ (the Maréchal criterion). We repeated this measurement 6 times. The mean RMS of the differential wavefront was $0.022 \mu\text{m}$, again well below the diffraction limit of $0.056 \mu\text{m}$. The RMS of the wavefront measured with and without the ND filters are shown in Supplementary Figure 9. The RMS of the wavefront measured with the ND filters was $0.309 \pm 0.004 \mu\text{m}$ (Mean \pm SD, $n = 6$) and the RMS of the wavefront measured without the ND filters was $0.311 \pm 0.003 \mu\text{m}$ (Mean \pm SD, $n = 6$). Two-sample t-test does not reject the null hypothesis that the mean of the RMS measured with and without ND filters are the same ($p = 0.39$), at the 5% significance level even if equal variances are not assumed. Based on these results, we conclude that the ND filters have a negligible effect on the measured wavefront.

Supplementary Figure 9. RMS of the wavefront measured with and without the ND filters for six repeated measurements.”

22. Lines 656-657 – The procedure for correcting for the subject’s sphero-cylindrical error should be expanded here. Was this done based on the subject’s prescription? Empirically based on the AO wavefront sensor? If done empirically, was this an objective measure (minimize RMS error)? Or subjective (sharpest spots)?

Response: Thank you for this suggestion. We need only a coarse estimate of the subject's spherocylindrical error, which we apply to the DM before closing the AO loop to bring the subject within the dynamic range of the wavefront sensor. Once within the range, the AO control takes over. The subject's prescription works well for this purpose, but if it is not available, we empirically adjust the sphere and cylinder correction with the DM until the wavefront RMS error is nominally minimized.

We have added the above information to the Methods section on Page 22 of the revised manuscript:

“We initially corrected the subject's spherocylindrical refractive error using the subject's prescription and the DM, before closing the AO loop to bring the subject within the dynamic range of the wavefront sensor. Once within the range, the AO control takes over. If the subject's prescription is unavailable, we empirically adjust the sphere and cylinder corrections with the DM until the RMS wavefront error is minimized.”

Supplemental

23. Figure S1 – The authors only briefly discuss the treatment of their camera with rolling shutter as a global shutter. More generally, are there significant differences between expected outcomes using a rolling vs. global shutter? Is one preferable? What are the tradeoffs?

Response: Thank you for asking this important question. We have added a new Supplementary Note in the supplementary material to discuss this topic:

“Supplementary Note 1: Comparing rolling shutter and global shutter modes for the SHWS camera in AO ophthalmoscopy

Rolling shutter and global shutter cameras use fundamentally different exposure schemes for capturing images. In global shutter mode, all pixels begin exposing at the same time, capturing the entire image at once. By contrast in rolling shutter mode, adjacent rows of pixels begin exposing at slightly different times, capturing the image line by line. This staggered exposure can lead to distortion in the image for objects or scenes that are moving faster than the rolling shutter speed.

A key advantage of global shutter cameras is that they do not suffer from such image distortion. However, they have a number of disadvantages compared to rolling shutter cameras that may outweigh this advantage.

1. **Lower frame rate:** Global shutter cameras require a reference frame to be read out from the sensor in addition to the signal frame. As a result, the maximum frame rate of global shutter is only half that of rolling shutter cameras [3]. This lower frame rate leads to a reduced AO loop rate, which reduces AO bandwidth and power rejection performance.

2. **Increased noise:** Global shutter cameras generally exhibit higher noise levels. Their RMS readout noise is at least 1.41 times greater than that of rolling shutter cameras, due to the additional frame mentioned in (1). This results in lower sensitivity.

3. **Lower quantum efficiency:** Global shutter cameras often have lower quantum efficiency because their pixel architecture includes extra electronics for the shuttering mechanism, which reduces the effective area available for light collection and results in a lower fill factor. Note that the quantum efficiency of a camera includes non-material related loss factors such as the fill factor [4].

4. **Smaller market:** The market for global shutter cameras is smaller with fewer options available. This is particularly true for scientific CMOS (sCMOS) cameras, which are attractive for wavefront

sensing in the eye because of their high sensitivity, speed, dynamic range and field of view. For example, sCMOS cameras sold by Hamamatsu (the manufacturer of the SHWS camera used in our study) are only available with rolling shutters.

5. Higher cost: Global shutter cameras are generally more expensive due to their more complex pixel architecture and circuitry, as well as their smaller market size.

Which shutter mode is better depends on the specific application. For AO ophthalmoscopy, rolling shutter mode allows up to twice the AO loop rate compared to global shutter mode, improving power rejection performance and increasing AO bandwidth. In addition, the reduced noise and higher quantum efficiency of rolling shutter mode improve the centroiding accuracy of the SHWS. However, the primary concern with rolling shutter mode is the potential for image distortion affecting wavefront aberration measurements.

Using the parameters of our Hamamatsu Lightning camera, we consider this concern and find that image distortion caused by the rolling shutter is largely inconsequential for AO ophthalmoscopy. For our rolling shutter camera, the difference in exposure start time between the top and the bottom rows of pixels in our region of interest (1920 W × 1840 H pixels) is 2.9 ms, limiting the maximum frame rate of the camera to 342 Hz. Hence, for aberrations that change more slowly than 2.9 ms (or 342 Hz), image distortion is minimal and will not impact aberration measurements. Aberrations that change faster than 2.9 ms will experience some distortion, but these aberrations are exceedingly small. Based on our measurements of ocular aberration power spectra (Supplementary Figure 2) and the assumption that ocular aberrations follow a power law, the portion of the aberrations changing faster than 2.9 ms (or 342 Hz) is <0.1% of the total aberration power.

The requirement for correcting aberrations is actually less stringent than the requirement for measuring them (discussed in the last paragraph). This is because the goal of AO is to minimize the displacements of the focal spots formed by the SHWS lenslets (sub-apertures). Therefore, minimizing the displacement of a single focal spot depends only on those rows of pixels associated with that specific lenslet. Our SHWS has 20 rows of lenslets, so the distortion for each focal spot image is minimal for aberrations that change more slowly than $\sim 2.9 \text{ ms}/20 = 0.145 \text{ ms}$ (or 6.9 kHz). Based on our measurements of the ocular aberration power spectra (Supplementary Figure 2) and the assumption that ocular aberrations follow a power law, we expect that aberrations changing faster than 0.145 ms (or 6.9 kHz) contribute extremely little (<0.01% of the total power). Even if such aberrations were significant and caused distortion with rolling shutter, their frequency (6.9 kHz) is so much higher than the bandwidth of our AO system (38 Hz, the maximal frequency our AO system can correct). Thus, the AO system would not correct these aberrations, whether distorted or not.

This analysis, based on the specific parameters of our wavefront sensor camera, can also be applied to other cameras.”

24. Supplementary Note 1 – While some of the transfer functions have fairly obvious forms (zero-order hold and delay), others are not as obvious to me (e.g. HWFS). It would be helpful to describe the origins of these equations or cite appropriate studies using this model.

Response: Thank you for this suggestion. We have added citations [5-8] at the end of the sentence, “Expressions for the five transfer functions in Supplementary Figure 4 are introduced below”.

25. Similarly, the choice of integrator controller is different than what I’ve seen in the past for theoretical analyses of AO control in retinal imaging systems (typically modeled as (loop gain)/s).

Where does the Hcontroller expression derived here come from? How does this choice of controller affect the results vs. other choices?

Response: $H_{\text{controller}} = (\text{loop gain})/s$ is applicable for continuous-time signals. However, in practice, the AO control system operates with discrete-time signals at a specific sampling rate. As the controller works in the sampled time domain, its transfer function is defined by the Z-transform [1,2]. Considering discrete time with the sampling interval being T_{hold} , the transfer function of the integrator is expressed using the z-transform as $H(z) = (\text{loop gain})/(1-z^{-1})$ [1, 2]. When we do frequency analysis, we let $z = \exp(sT_{\text{hold}})$. Then, we obtained the equation we used for the integrator: $H_{\text{controller}} = (\text{loop gain})/[1-\exp(-sT_{\text{hold}})]$, which is also shown in [1,2].

The expression of the integrator controller for the discrete time domain takes into account a finite sampling rate ($1/T_{\text{hold}}$), which is more appropriate for modeling a practical AO system. We have added these citations after the sentence “Expressions for the five transfer functions in Supplementary Figure 4 are introduced below”.

References:

[1]. F. Roddier, Adaptive Optics in Astronomy (Cambridge University Press, Cambridge, 1999).

[2]. H. Hofer, L. Chen, G. Y. Yoon, B. Singer, Y. Yamauchi, and D. R. Williams, "Improvement in retinal image quality with dynamic correction of the eye's aberrations," Opt. Express 8, 631-643 (2001).

26. Lines 146-157 – What was the loop gain value used here?

Response: The loop gain value was 1. We have added this information on Page 9 of the revised Supplementary Materials.

27. Line 264 – Similar to comment 20 above, instead of specifying the minimal AO loop rate, it may be more general to specify this as the rate-gain product.

Response: Thank you for this suggestion. As explained in our response to comment 20 above, we prefer to use the AO loop rate in this particular context for the Fig. 5b and 5c results, because the power rejection curve of an AO system not only depends on the rate-gain product, but also depends on three other timing parameters, two of which are related to the AO loop rate and are independent of the AO loop gain, while the other is a constant.

Nevertheless, we agree with the reviewer that the method presented in Supplementary Note 5 and Equation (M5) are general and can be readily used to quantitatively study the effect of the rate-gain product on the temporal error, provided that all timing parameters that influence the AO's power rejection curve are known.

Response to review comments

Reviewer #1 (Remarks to the Author):

The authors have done an excellent job in revising the manuscript and have carefully addressed all of the questions and comments raised. As a result the manuscript is stronger and will be even more beneficial for the optics community.

Response: We thank the reviewer for the positive feedback on our revised manuscript and for their critical reading of it.

Reviewer #2 (Remarks to the Author):

The authors have adequately addressed all concerns and suggestions. I propose that the manuscript shall be accepted with the implemented edits.

Response: We thank the reviewer for the positive feedback on our revised manuscript and for their critical reading of it.

Reviewer #3 (Remarks to the Author):

The authors have done excellent work in responding to my and the other reviewer's points. The revised manuscript includes more supporting data across more clinically significant imaging scenarios which further supports the key points of the manuscript. Overall, the revised manuscript is very strong and clearly addresses all of the concerns raised.

Response: We thank the reviewer for the positive feedback on our revised manuscript and for their critical reading of it.